# Pantothenate kinase 4 controls skeletal muscle substrate metabolism

Adriana Miranda-Cervantes [1,2], Andreas M. Fritzen [3,4], Steffen H. Raun[3,4], Ondřej Hodek [5,6], Lisbeth L. V. Møller [3,4], Kornelia Johann[1,2], Luisa Deisen[1], Paul Gregorevic [7,8], Anders Gudiksen[9], Anna Artati[10], Jerzy Adamski [11,12,13], Nicoline R. Andersen[3], Casper M. Sigvardsen [3], Christian S. Carl [3], Christian T. Voldstedlund [3], Rasmus Kjøbsted [3], Stefanie M. Hauck [2,10], Peter Schjerling[14,15], Thomas E. Jensen [3], Alberto Cebrian-Serrano [2,16], Markus Jähnert [2,17], Pascal Gottmann [2,17], Ingo Burtscher[2,18], Heiko Lickert [2,18,19], Henriette Pilegaard[9], Annette Schürmann [2,17,20], Matthias H. Tschöp [2,16,19], Thomas Moritz [5,21], Timo D. Müller [2,16,22], Lykke Sylow [3,4], Bente Kiens [3], Erik A. Richter [3] & Maximilian Kleinert [1,2,3,16,20] ✉

Metabolic flexibility in skeletal muscle is essential for maintaining healthy glucose and lipid metabolism, and its dysfunction is closely linked to metabolic diseases. Exercise enhances metabolic flexibility, making it an important tool for discovering mechanisms that promote metabolic health. Here we show that pantothenate kinase 4 (PanK4) is a new conserved exercise target with high abundance in muscle. Muscle-specific deletion of PanK4 impairs fatty acid oxidation which is related to higher intramuscular acetyl-CoA and malonyl-CoA levels. Elevated acetyl-CoA levels persist regardless of feeding state and are associated with whole-body glucose intolerance, reduced insulin-stimulated glucose uptake in glycolytic muscle, and impaired glucose uptake during exercise. Conversely, increasing PanK4 levels in glycolytic muscle lowers acetyl-CoA and enhances glucose uptake. Our findings highlight PanK4 as an important regulator of acetyl-CoA levels, playing a key role in both muscle lipid and glucose metabolism.

Skeletal muscle (SkM) makes up approximately 30-40% of body mass[1] and plays a crucial role in maintaining overall glucose and lipid homeostasis. SkM is responsible for the majority of glucose disposal stimulated by insulin[2] and serves as the primary storage site for glucose in the form of glycogen[3]. Metabolic conditions that impair the ability of SkM to efficiently use or store glucose pose a significant challenge to maintaining overall glucose homeostasis in the body.

Obesity is associated with insulin resistance of SkM, which is considered a primary risk factor for the development of type 2 diabetes (T2D)[4]. Obesity-induced insulin resistance in SkM involves complex mechanisms, including inflammation, mitochondrial

dysfunction, endoplasmic reticulum stress, and lipotoxicity[5–7]. In obesity, adipocytes exceed their lipid storage capacity, releasing fatty acids and cytokines into the bloodstream. This surplus of lipids accumulates in non-adipose tissues, causing local inflammation and contributing to insulin resistance development. In addition, it has been suggested that obesity induces metabolic inflexibility in SkM[8]. Normally, we experience alternating periods of feeding and fasting, necessitating metabolic flexibility to switch between using glucose and fatty acids as energy sources. The reciprocal relationship between glucose and fatty acid was first described 60 years ago[9], but the underpinning mechanism are not fully elucidated. With prolonged and

frequent overeating, an excess supply of glucose and fatty acids to SkM cells causes mitochondrial "congestion"[10], derailing the coordinated uptake of glucose and oxidation of fatty acids. This metabolic perturbation is thought to contribute to insulin resistance in SkM and challenges systemic glucose homeostasis.

Exercise has been found to increase glucose uptake in SkM in both healthy individuals and in those with insulin resistance[11], making it an effective intervention for reducing blood glucose levels in people with T2D. Furthermore, exercise has been demonstrated to improve the metabolic flexibility of SkM[8] and enhance insulin sensitivity[12], thereby contributing to better overall glucose balance in the body. Thus, exercise-associated signaling mechanisms include potential therapeutic targets that regulate efficient utilization of glucose and lipids, some of which may still be unidentified. We here show that pantothenate kinase 4 (PanK4) is a new exercise target abundant in both rodent and human SkM. Moreover, we demonstrate that PanK4 plays a role in coordinating the regulation of glucose and lipid metabolism in SkM.

## Results

### Pantothenate kinase 4 (PanK4) is a new exercise target in skeletal muscle

Datasets on exercise- or contraction-induced changes in protein phosphorylation[13,14] indicate that pantothenate kinase 4 (PanK4) has a conserved phosphorylation site at Ser63 (p-PanK4$^{Ser63}$) that is acutely increased by exercise or muscle contractions in rodent and human SkM (Fig. 1A). PanK4 belongs to the family of pantothenate kinases (PanK1-4) that regulate the biosynthesis of coenzyme A (CoA), a pivotal coenzyme required in many cellular processes, including the synthesis, storage, and uptake of fatty acids (FAs)[15]. It has recently become evident that PanK4, however, does not possess classical pantothenate kinase activity[16]. Instead PanK4 may rather antagonize the production of CoA by acting as a phosphatase[17]. This potential antagonistic action is consistent with the pantothenate kinase domain sequence of PanK4 being significantly different from that of its family members, PanK1-3 (Fig. S1A). Clearly, the biological

function of PanK4 is unique from its family members and its metabolic and physiological function in mature insulin-responsive organs has not been investigated.

We generated and validated a phospho-specific antibody against p-PanK4$^{Ser63}$ (Fig. S1B, C). Using this antibody, we observed that after vigorous cycling exercise in humans (~10 min at 77–88% $W_{max}$), p-PanK4$^{Ser63}$ increased 2.5-fold in vastus lateralis muscle and returned to pre-exercise levels five hours later (Fig. 1B). Similarly, p-PanK4$^{Ser63}$ increased in SkM following prolonged endurance (90 min of continuous cycling at 60% of $VO_2$ peak), sprint (three bouts of 30-s all-out cycling), and strength (6 sets of 10 repetitions of bilateral knee extensions) exercise, with similar peak levels (~4-fold) with all three exercise modalities (Fig. 1C). These data indicate that PanK4 is an exercise target in human SkM, responding to different types of exercise. Recently, Thr406 was identified as a insulin-responsive site on PanK4 in cultured cancer cells[17]. Phosphoproteomic analyses indicate that this site also responds to insulin but not to exercise in human SkM (Fig. S1D).

### PanK4 is abundant in striated muscle and its expression associates with conditions of metabolic dysregulation

We next explored where PanK4 is expressed. We detected high levels of *Pank4* mRNA in skeletal and cardiac muscle, hypothalamus, and the pituitary gland of mice (Fig. 1D). In mouse SkM, *Pank4* is the most abundant *Pank* isoform (Fig. 1E). Protein levels of PanK4 are similar among different SkM but generally higher than in other organs such as adipose tissues and liver (Fig. 1f). Conversely, protein levels of PanK1 and PanK2 are higher in fat and liver than in SkM, with the exception of PanK2 being abundant in oxidative, slow-twitch soleus muscles (Fig. 1F). Murine *Pank4* mRNA abundance was decreased in SkM from mice subjected to an obesogenic 20 weeks high-fat diet (Fig. S1E)[18]. We identified a human *PANK4* variant (rs7535528) that is linked to glycated hemoglobin (HbA$_{1c}$)[19] and body mass index (BMI)[20] traits (Fig. S1F), implying a potential translational significance for PanK4 in the regulation of glycemic control and energy balance. Collectively, these findings highlight a significant abundance of PanK4 in SkM and point

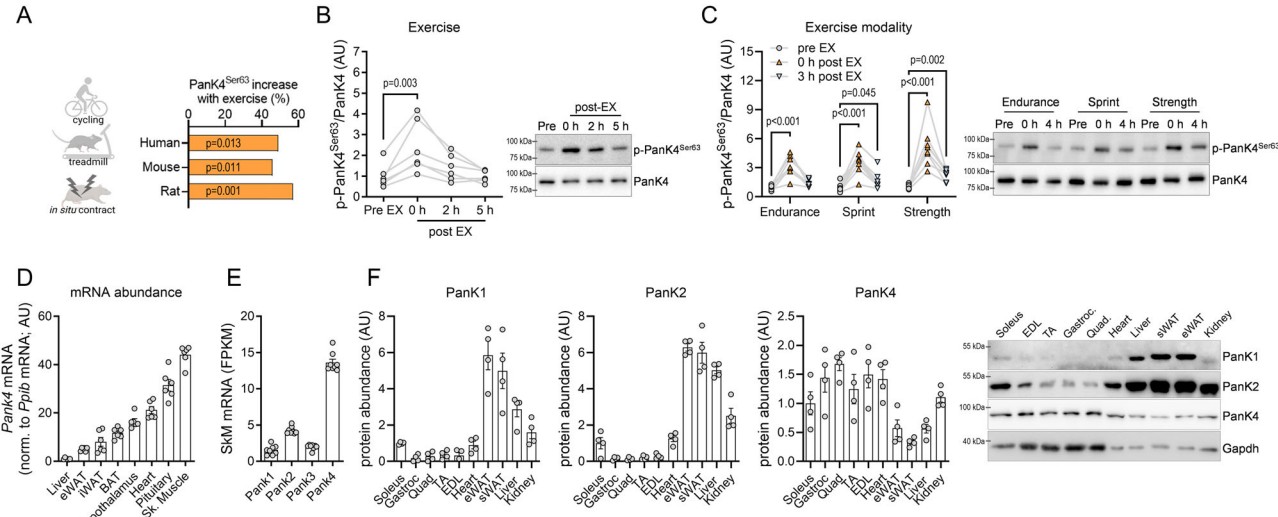

**Fig. 1 | PanK4 is an exercise target abundant in skeletal muscle and associated with metabolic dysregulation.** **A** PanK4 Ser63 phosphorylation (p-PanK4$^{Ser63}$) and its regulation by exercise or muscle contraction in skeletal muscle from humans and rodents indicated by phosphoproteomics[13,14]. p-PanK4$^{Ser63}$ in human vastus lateralis before (Pre) and post-exercise indicated time-points into recovery from vigorous cycling exercise (**B**, $n = 6$), prolonged endurance cycling, sprint cycling and strength exercises (**C**, $n = 8$). **D** *Pank4* mRNA in indicated male mouse tissues ($n = 6$). **E** mRNA abundance of indicated *Pank* isoforms determined by whole genome transcriptome analysis in male mouse skeletal muscle ($n = 8$). **F** abundance

of PanK1, PanK2 and PanK4 protein in indicated tissues from male mice and representative Western blots ($n = 4$). **D–F** data are presented as mean values +/− SEM. Statistic: **A** significantly regulated phosphopeptides (adjusted $P$-values less than 0.05) were identified in larger data sets using LIMMA's moderated t-statistics with the eBayes method (see Refs. 13,14 for more detail). **B** repeated measures one-way ANOVA and **C** repeated measures two-way ANOVA with log2-transformed data and Šidák post hoc testing. Source data are provided as a Source Data file. AU = arbitrary units. Schematic in Fig. 1A was created in BioRender. Kleinert, M. (2024) BioRender.com/t99l144.

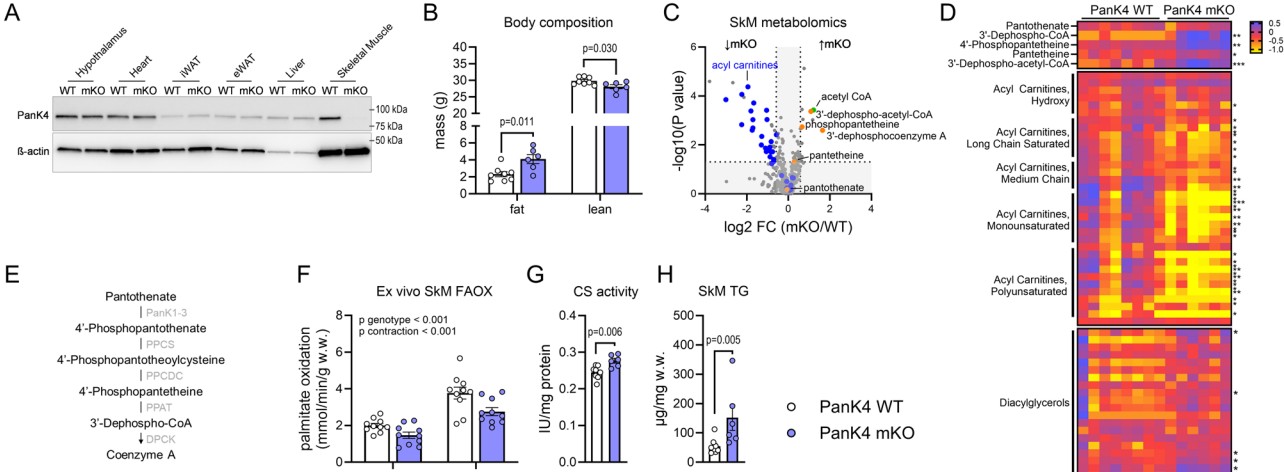

**Fig. 2 | Skeletal Muscle PanK4 regulates fatty acid oxidation. A** PanK4 protein abundance in indicated tissues from male PanK4 wildtype (WT) and muscle-specific PanK4 knockout (mKO) mice at age 28 weeks. **B** body composition at age 26 weeks ($n = 8$ for WT and $n = 6$ for mKO) of male PanK4 WT and PanK4 mKO mice. **C, D**, volcano plot of 508 detected metabolites and heatmap of indicated metabolites in TA muscle from male glucose-stimulated PanK4 WT and PanK4 mKO mice at age 28 weeks ($n = 8$ for WT and $n = 6$ for mKO). **E** overview of the Coenzyme A biosynthesis pathway. **F** palmitate oxidation in ex vivo basal or contracted soleus muscles from ad libitum fed male PanK4 WT and PanK4 mKO mice at age 12-19 weeks ($n = 10$). Citrate synthase activity and muscle triglyceride (TG)

concentrations in soleus (**G**) and gastrocnemius (**H**) muscles from male glucose-stimulated PanK4 WT and PanK4 mKO mice at age 28 weeks ($n = 8$ for WT and $n = 6$ for mKO). Statistic: **B** two-tailed unpaired t-test within fat or lean mass; **C, D** two-tailed welch test; **F** two-way ANOVA; **G, H**, two-tailed unpaired t-test. Statistical analyses were conducted with log2-transformed data. **B, F–H**, data are presented as mean values +/− SEM. PanK1-3 pantothenate kinase 1-3, PPCS phosphopantothenate-cysteine ligase, PPCDC phosphopantothenoylcysteine decarboxylase, PPAT phosphopantetheine adenylyltransferase, DPCK = 3'-Dephospho-CoA kinase. Source data are provided as a Source Data file.

to a potential involvement of PanK4 in the regulation of body mass and glucose metabolism.

## Germline deletion of PanK4 reduces circulating IGF-1 and stunts growth

To investigate the metabolic function of PanK4, we first generated whole-body PanK4 knockout (PanK4 KO) mice by replacing the open reading frame of *Pank4* by a NLS-lacZ-2A-H2B-Venus cassette[21]. Chow-fed male and female PanK4 KO mice exhibited reduced SkM mass (Fig. S2C, D). Female PanK4 KO mice also had lower body weight, smaller hearts and livers, were overall shorter and had reduced levels of circulating insulin-like growth factor 1 (IGF-1) (Fig. S2B, D, F). Male PanK4 KO mice had similar body weight and length as their WT counterparts, but their heart mass tended ($p < 0.1$) to be lower and circulating IGF-1 levels were also reduced (Fig S2A, C, E). These data indicate that PanK4 is critical for organismal development, with more pronounced effects in female mice.

## PanK4 regulates the availability of fatty acyl-carnitines in SkM

Given the pleiotropic effects of IGF-1 on SkM growth and metabolism, the lower IGF-1 levels in the whole-body PanK4 KO mice confounded our questions regarding the role of PanK4 in SkM. We therefore generated SkM-specific PanK4 knockout mice using the Cre-loxP system with SkM-specific Cre recombinase expression, controlled by the human α-skeletal actin promoter: $Pank4^{flox/flox}{:}HSA\text{-}Cre^{-/+}$ (PanK4 mKO) (Fig. 2A). PanK4 mKO mice had normal body weight and food intake (Fig. S3A, B), but had twice the amount of fat mass compared to control $Pank4^{flox/flox}{:}HSA\text{-}Cre^{-/-}$ (PanK4 WT) mice (Fig. 2B). This was associated with 46% and 70% larger epididymal and inguinal white adipose tissue depots, respectively (Fig. S3C). Conversely, PanK4 mKO mice had two grams less fat-free mass (Fig. 2B); however, this was not accompanied by changes in heart, SkM, BAT, and liver mass (Fig. S3C). Total energy expenditure and whole-body substrate utilization measured by indirect calorimetry were normal in PanK4 mKO mice, regardless of whether they were fed standard chow, subjected to an overnight fast, or challenged with high-fat diet (Fig. S3D, E, G, H).

Notably, PanK4 mKO mice showed greater locomotor activity than PanK4 WT mice on a chow diet, but this effect was absent on a high-fat diet (Fig. S3F, I, J).

To gain insights into potential metabolic function of PanK4 specifically in SkM, we applied nontargeted metabolomics (supplementary data 1). We found that several intermediate metabolites of the CoA synthesis pathway (e.g., 4'-Phosphopantetheine and 3'-Dephospho-CoA) were increased in PanK4 mKO muscles (Fig. 2C–E & Fig. S4A), supporting recent findings in cancer cells that PanK4 might act as a potential repressor of the CoA pathway[17]. Strikingly, most fatty acyl-carnitines were lower in the SkM of PanK4 mKO mice (Fig. 2C, D & Fig. S4A). Among the 34 fatty acyl-carnitine species detected, 24 were significantly lower in the tibialis anterior (TA) muscles of PanK4 mKO mice, with the remaining 10 unchanged (Fig. 2C, D). Similarly, in the gastrocnemius muscle, 20 acyl-carnitines were lower and the remaining 14 were unchanged in PanK4 mKO mice (Fig. S4A). Levels of intramuscular free carnitine and fatty acids (i.e., acyl groups) were unchanged (Fig. S4B, C), and circulating fatty acids were normal (Fig. S4D), indicating that the supply of precursor metabolites does not account for the significant SkM fatty acyl-carnitine imbalance.

## PanK4 regulates fatty acid oxidation in SkM

We hypothesized that lower levels of fatty acyl-carnitine reflect a deficiency in fatty acid entry into the mitochondria, resulting in decreased beta-oxidation. To investigate the latter, we examined fatty acid oxidation (FAOX) using radiolabeled palmitate in ex vivo incubated soleus muscle. We focused on the oxidative soleus muscle, because of its high capacity for FAOX[22]. We examined both basal- and contraction-stimulated FAOX, with contraction being a potent physiological stimulus of FAOX in SkM. Contraction-induced FAOX was 2-fold higher than basal. Strikingly, both basal- and contraction-induced FAOX were 25% decreased in PanK4 mKO muscles (Fig. 2F). We next asked whether a decrease in mitochondrial content explained this defect in FAOX in PanK4 mKO mice. Mitochondrial content correlates with the activity of citrate synthase in muscle cells[23]. SkM citrate synthase activity was moderately increased in

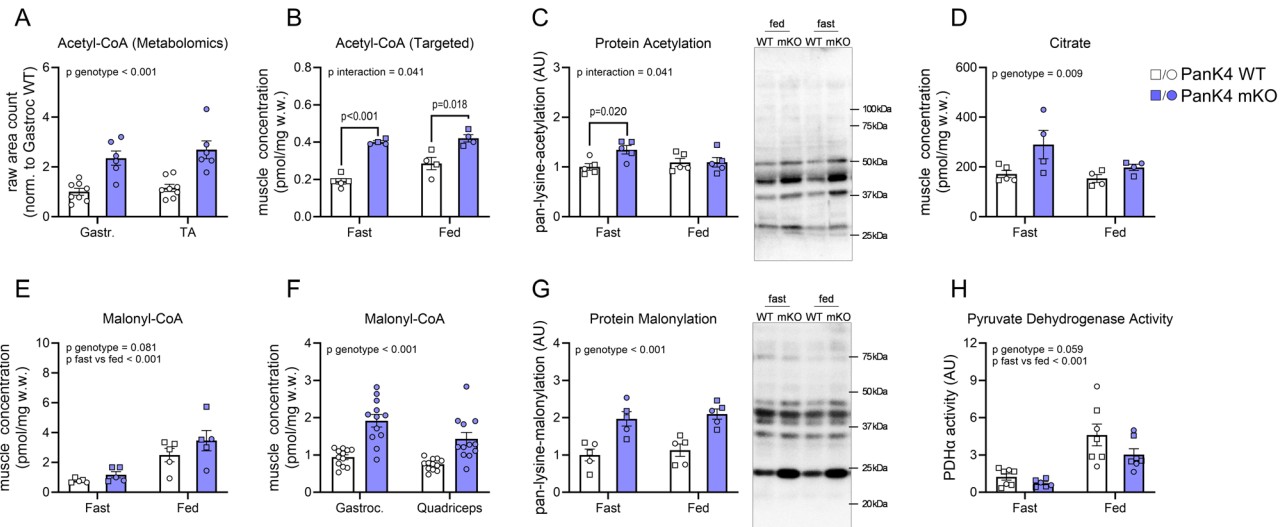

**Fig. 3 | PanK4 regulates acetyl-CoA and metabolic flexibility in skeletal muscle.**
**A** acetyl-CoA determined by non-targeted metabolomics in indicated skeletal muscle from male PanK4 WT and PanK4 mKO mice weeks (*n* = 8 for WT and *n* = 6 for mKO). **B** targeted analysis of acetyl-CoA in gastrocnemius muscle from fasted and fed male (circles) and female (squares) PanK4 WT and PanK4 mKO mice (*n* = 5 for fast/WT, *n* = 4 for fed/WT, *n* = 4 for fast/mKO *n* = 4 for fed/mKO). **C** representative Western blots and quantification of pan-lysine acetylation in gastrocnemius muscle from male (circles) and female (squares) PanK4 WT and PanK4 mKO mice (*n* = 5). **D–F** targeted analysis of citrate and malonyl-CoA in gastrocnemius muscle from fasted and fed male (circles) and female (squares) (**E**, *n* = 5 for fast/WT, *n* = 4 for fed/WT, *n* = 4 for fast/mKO *n* = 4 for fed/mKO; **F**, *n* = 5) and in

indicated muscle from fed male (**G**, *n* = 12) PanK4 WT and PanK4 mKO mice.
**G** representative Western blots and quantification of pan-lysine malonylation in gastrocnemius muscle from male (circles) and female (squares) PanK4 WT and PanK4 mKO mice (*n* = 5). **H** activity of pyruvate dehydrogenase in the active form (PDHa) activity in gastrocnemius muscle from male (circles) and female (squares) PanK4 WT and PanK4 mKO mice (*n* = 7 for fast/WT, *n* = 7 for fed/WT, *n* = 6 for fast/mKO *n* = 7 for fed/mKO). **A–H** data are presented as mean values +/- SEM. Statistic: **A–H**, two-way ANOVA. Statistical analyses were conducted with log2-transformed data. Šidák post hoc testing was performed. Source data are provided as a Source Data file. AU = arbitrary units.

PanK4 mKO mice (Fig. 2G), suggesting that mitochondrial content is unlikely to explain the defects in FAOX. This is supported by similar protein abundance of key electron transport chain proteins (Fig. S4E). Moreover, mRNA levels of genes encoding for proteins involved in fatty acid entry into muscle cells (*Fabp3*, *Fabp4*, *Cd36*, and *Fatp4*), as well as those related to the carnitine shuttle (*Cpt1b* and *Cpt2*) facilitating the mitochondrial entry of fatty acids, were unaltered in PanK4 mKO muscles (Fig. S4F). It remains possible that the activity of these proteins is altered. Notably, the reduction in FAOX coincided with an increase in diacylglycerol (DAG) species (Figs. 2D and S4G), along with a three-fold elevation in intramuscular triacylglycerol (TG) concentration (Fig. 2H) in SkM from PanK4 mKO mice. We surmised that instead of undergoing oxidation, fatty acids are shunted towards storage pathways. Collectively, these data indicate that PanK4 controls fatty acid utilization in oxidative SkM.

### PanK4 regulates acetyl-CoA and metabolic flexibility in SkM
To investigate how PanK4 regulates lipid metabolism in SkM, we revisited the metabolomics data. We were intrigued by a ~100% increase in SkM acetyl-CoA in PanK4 mKO mice (Fig. 3A) because acetyl-CoA levels serve as a critical gauge of cellular fuel status, regulating the flux of glucose and fatty acids[24]. We could confirm elevated SkM acetyl-CoA levels in PanK4 mKO mice both during fasted and fed conditions by targeted analysis (Figs. 3B and S5A, B). This was associated with an increase in global protein acetylation in SkM from PanK4 mKO mice (Fig. 3C), consistent with the notion that elevated acetyl-CoA per se is sufficient to affect the acetylproteome[24]. Free CoA levels remained unchanged in PanK4 mKO muscle (Fig. S5C), suggesting that despite the potential role of PanK4 as a repressor of CoA biosynthesis[17], free CoA homeostasis is largely maintained in SkM in the absence of PanK4. The levels of mRNAs encoding for PanK1-3, which could potentially compensate free CoA levels, were unaltered in PanK4 mKO SkM (Fig. S5D).

Elevated mitochondrial acetyl-CoA levels can negatively impact cellular energy metabolism, by leading to the accumulation of citrate, which can be exported from the mitochondria to the cytosol. In the cytosol, high levels of citrate can lead to increased levels of acetyl-CoA and malonyl-CoA. The latter, malonyl-CoA, is a potent allosteric inhibitor of CPT1, a key component of the carnitine shuttle that facilitates mitochondrial import of fatty acids[25,26]. In agreement with a potential spillover effect due to high acetyl-CoA concentration, we found elevated levels of citrate and malonyl-CoA in SkM of PanK4 mKO mice compared to control mice (Fig. 3D–F), and these higher malonyl-CoA levels were associated with increased global protein malonylation in SkM (Fig. 3G).

Typically acetyl-CoA levels are thought to be more important for the fine-tuning of fatty acid utilization (via e.g. malonyl-CoA) than for regulating glucose metabolism, however, they have also been implicated in the regulation of glucose metabolism by allosterically inhibiting pyruvate dehydrogenase (PDH)[27], an enzyme that facilitates the conversion of pyruvate to acetyl-CoA (Fig. 3D). This link might be relevant under conditions of persistently elevated acetyl-CoA levels as is the case in SkM from PanK4 mKO mice. In accordance with a potential negative feedback loop being activated, we observed a trend (*p* = 0.059) towards reduced activity of PDH in the active form (PDHa) in PanK4 mKO SkM (Fig. 3H). Intriguingly, this reduced activity was not explained by an altered phosphorylation status of PDH (Fig. S5E–L), implying that other post-translational modifications (e.g., acetylation or succinylation) of PDH might play a role. Overall, our findings indicate that high levels of acetyl-CoA lead to an increase in malonyl-CoA concentration which could impair FAOX by decreasing the carnitine shuttle-mediated entry of long-chain fatty acids into mitochondria. We also observed a trend for regulation of PDH indicating that SkM glucose metabolism might also be affected by the absence of PanK4.

Therefore, we evaluated glycemic outcomes in PanK4 mKO mice. On a whole-body level, glucose tolerance was impaired while fasting

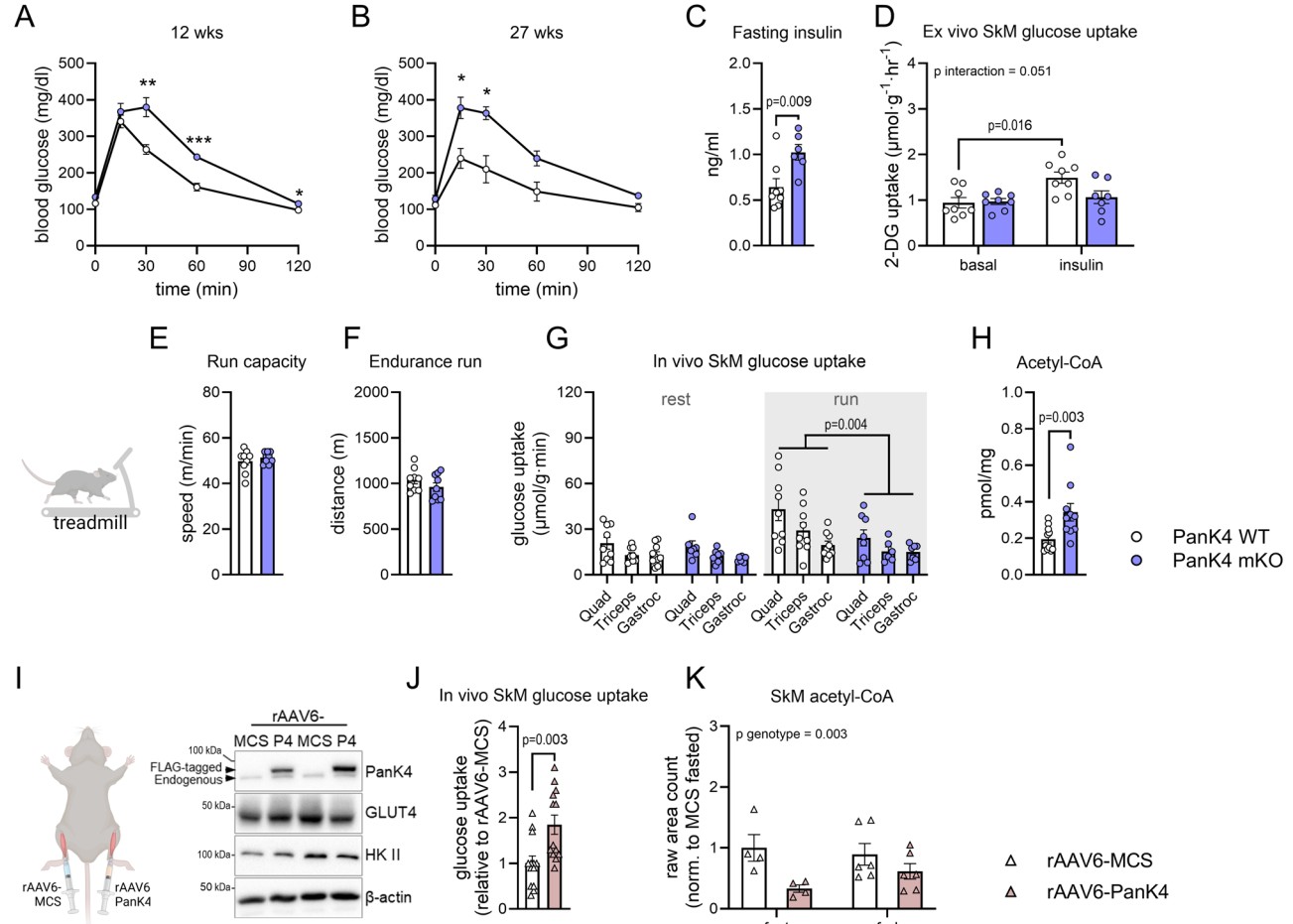

**Fig. 4 | Skeletal muscle PanK4 regulates whole-body insulin sensitivity and muscle glucose uptake. A, B** glucose tolerance in male PanK4 WT and PanK4 mKO mice at indicated ages ($n = 8$ for WT and $n = 6$ for mKO). **C** fasting plasma insulin concentrations in male PanK4 WT and PanK4 mKO mice ($n = 8$ for WT and $n = 6$ for mKO). **D** ex vivo basal- or insulin (3.0 nM)-stimulated glucose uptake in skeletal muscle (EDL) from male PanK4 WT and PanK4 mKO mice ($n = 8$ for WT/basal, $n = 8$ for WT/insulin, $n = 8$ for mKO/basal, $n = 7$ for mKO/insulin). Maximal running speed (**E**) and maximal distance (**F**) during treadmill running of male PanK4 WT and PanK4 mKO ($n = 9$). **G** glucose uptake into indicated skeletal muscle during 20 min of rest or treadmill running at 75% of maximal running capacity in male PanK4 WT and PanK4 mKO mice (rest: $n = 9$ for WT/quad, $n = 9$ for WT/triceps, $n = 10$ for WT/gastroc, $n = 7$ for mKO; running: $n = 9$ for WT, $n = 8$ for mKO/quad, $n = 6$ for mKO/triceps, $n = 7$ for mKO/gastroc). **H** acetyl-CoA levels determined by targeted analyses in quadriceps muscle from male PanK4 WT and PanK4 mKO mice that had ran on the treadmill ($n = 11$ for WT and $n = 10$ for mKO). Male C57BL/6J mice aged 12–16 weeks were treated with recombinant adeno-associated virus serotype 6

encoding PanK4 (rAAV6:PanK4) injected into tibialis anterior (TA) muscle, while contralateral TA was injected with rAAV6:MCS as a control: PanK4 protein abundance (**I**) and glucose uptake (**J**, $n = 13$) in TA muscles were determined 14 days later in fed mice. **K** acetyl-CoA detected by non-targeted metabolomics in fasted or refed male C57BL6/J mice overexpressing PanK4 in one TA and MCS in contralateral TA ($n = 4$ for MCS/fast, $n = 6$ for MCS/fed, $n = 4$ for PanK4/fast, $n = 6$ for PanK4/fed). **A–H, J, K** data are presented as mean values +/− SEM. Statistic: **A, B** repeated measures two-way (time x genotype) ANOVA; **C** two-tailed student's $t$-test; **D** two-way (genotype x insulin) ANOVA; **G** two-way ANOVA (genotype x muscle) within rest or run; **H** two-tailed student's t-test. **J** two-tailed student's $t$-test; **K** two-way (genotype x feeding) ANOVA. Statistical analyses were conducted with log2-transformed data. Šidák *post hoc* testing. Source data are provided as a Source Data file. Schematic in **A** was created in BioRender. Kleinert, M. (2024) BioRender.com/m45y371; schematic in **I** was created in BioRender. Kleinert, M. (2024) BioRender.com/l81e607.

insulin levels were elevated in PanK4 mKO mice (Fig. 4A–C). Insulin-stimulated glucose uptake was similar to controls in PanK4 mKO oxidative soleus muscles (Fig. S6A) but impaired in more glycolytic extensor digitorum longus (EDL) muscles ex vivo (Fig. 4D). This impaired glucose uptake in EDL muscle is uncoupled from proximal insulin signaling (Fig. S6B–H). Like insulin, exercise is a potent physiological stimulator of SkM glucose uptake[3]. While absence of SkM PanK4 did not affect maximal running speed (Fig. 4E) or endurance capacity (Fig. 4F), exercise-induced SkM glucose uptake was impaired in PanK4 mKO mice (Fig. 4G). This coincided with higher post-exercise blood glucose levels in PanK4 mKO mice (Fig. S7A). Blood lactate increased with exercise but was not different between genotypes (Fig. S7B). Following exercise, SkM acetyl-CoA (Fig. 4H) and citrate (Fig. S7C) concentrations were higher in PanK4 mKO mice than in control mice. Exercise increased SkM p-AMPK[Thr172] to a similar extent in

PanK4 WT and PanK4 mKO mice, whereas p-PanK4[Ser63] trended to increase ($p = 0.051$) in SkM with exercise in PanK4 WT but not in PanK4 mKO mice (Fig. S7D–H). Notably, PanK4 mKO mice had approximately 100% higher SkM glycogen levels than WT controls, with an overall trend ($p = 0.052$) toward reduced glycogen levels following exercise (Fig S7I). Both in situ and ex vivo contractions increase p-PanK4[Ser63] in SkM, suggesting that contraction per se contribute to exercise-induced p-PanK4[Ser63], via mechanisms that likely involve calcium-dependent signaling (Fig. S8).

Acetyl-CoA is primarily derived from catabolism of protein, fat, and glucose. With both lipid and glucose utilization compromised, we explored whether increased acetyl-CoA content is explained by enhanced catabolism of branched-chain amino acids (BCAAs). Whole-genome RNA-sequencing of SkM of PanK4 WT and PanK4 mKO mice revealed no significant alterations in genes encoding enzymes related

to BCAA catabolism (Fig. S9A–C; supplementary data 2). Additionally, the protein abundance of branched-chain α-keto acid dehydrogenase (BCKDH), which catalyzes the irreversible oxidative decarboxylation of branched-chain α-keto acids, was similar in SkM from PanK4 WT and PanK4 mKO mice; and phosphorylation of BCKDH-E1α at Ser293, which inactivates BCKDH[28], was unaltered in muscle lacking PanK4 (Fig. S9D, E). Our metabolomics data also showed no coordinated increase in metabolites related to amino acid breakdown that could drive acetyl-CoA levels. Overall, this indicates that the increase in acetyl-CoA in PanK4 mKO mice is not explained by the augmented breakdown of BCAAs.

We next tested if increasing PanK4 protein abundance benefits glucose uptake into SkM. Using recombinant adeno-associated virus-based vectors, serotype 6 (rAAV6) administered via intramuscular injections (Fig. 4I), we overexpressed flag-tagged PanK4 in the glycolytic TA muscle in mice. We first performed a dose-response experiment, assessing the relationship between vector titer and PanK4 protein levels (Fig. S10A) and selected a titer that resulted in a ~5-fold increase in PanK4 protein abundance for subsequent experiments. Overexpression of PanK4 had no effect on muscle mass (Fig. S10B, C). However, PanK4 overexpression increased glucose uptake into resting TA by 80% (Fig. 4J). SkM acetyl-CoA decreased when Pank4 was over-expressed (Fig. 4K, Fig. S10D–F, supplementary data 3). Collectively, these data suggest PanK4 could be a regulator of glucose uptake in glycolytically fiber type-dominated SkM.

Furthermore, utilizing the flag-tagged PanK4, we immunopurified PanK4 using both anti-flag and anti-PanK4 antibodies (Fig. S10G) and investigated binding partners by mass spectrometry, with pulldowns from PanK4 KO SkM serving as a negative control (Supplementary data 4). Among the binding partners identified through both anti-flag and anti-PanK4 pulldown, identification of 3-hydroxybutyrate dehydrogenase 1 (BDH1) (Fig. S10H–J) is intriguing, as BDH1 is required for conversion of ketone bodies, taken up from the circulation, into acetyl-CoA inside SkM. Moreover, it has recently been shown that SkM BDH1 optimizes FAOX efficiency[29]. While a possible connection between PanK4 and BDH1 is speculative and requires extensive testing, it highlights how these data are a hypothesis-generating resource.

## Discussion
We identify PanK4 as an exercise target that is abundant in SkM. We found that PanK4 regulates SkM acetyl-CoA levels, fatty acid oxidation capacity in oxidative muscle and glucose uptake in glycolytic muscle. We also show that increasing PanK4 enhances glucose uptake in glycolytic SkM. Our findings highlight PanK4 as a regulator of SkM energy substrate metabolism.

To discover new regulators of metabolism, we focused on exercise signaling because exercise is a potent physiological challenge to SkM metabolic flexibility. Our interest lay in identifying exercise signals that occur in mice, rats, and humans, as we believed that conserved targets would be particularly important for SkM metabolism. This led us to PanK4, a conserved protein with homologs found in animals, fungi, and plants[16]. The family of pantothenate kinases are known to phosphorylate pantothenate to form 4'-phosphopantothenate, the first and rate-limiting step in the CoA biosynthesis pathway. However, mammalian PanK4 lacks this ability and does not contribute to CoA biosynthesis[16,30], rendering it a pseudo-pantothenate kinase. PanK4 is also the only family member containing a C-terminal domain of unknown function 89 (DUF89), which can exist independently or, in the case of PanK4, be fused with other proteins[31–33]. It was reported that DUF89 possesses metal-dependent phosphatase activity, targeting phosphorylated metabolites[34]. Recent studies in cultured cells propose that PanK4 dephosphorylates 4'-phosphopantetheine, an intermediate metabolite in CoA biosynthesis, thereby antagonizing CoA production. Our data indicate that this mechanism may extend to SkM, as the absence of PanK4 led to increased levels of 4'-

phosphopantetheine. However, we observed no significant changes in SkM-free CoA levels, suggesting that free CoA homeostasis is maintained through alternative mechanisms in SkM. A limitation of our study is that our metabolite analysis provides only a snapshot and does not necessarily reflect flux, and free CoA levels in SkM may vary at other times of the day. Additionally, we cannot rule out changes in total CoA levels, which include free CoA and all CoA-containing metabolites (e.g., acetyl-CoA).

Our discovery of an exercise-responsive phosphorylation site on PanK4 complements the recent identification of an insulin-responsive site (T406) on PanK4[17]. However, the function of these phosphorylation sites on PanK4 remains unclear. It would be intriguing to understand whether they affect phosphatase activity of PanK4. Given that insulin is an anabolic stimulus, while exercise is a catabolic event, it raises the possibility that these two phosphorylation sites integrate competing cellular inputs, fine-tuning the function and activity of PanK4. Unraveling the regulation of PanK4 in SkM and other insulin-responsive organs will be of great interest in future research, especially with the link to acetyl-CoA. In liver, for example, acetyl-CoA can enter the TCA cycle, serve as a substrate in ketogenesis, or be carboxylated to yield malonyl-CoA for de novo lipogenesis.

We were surprised by the magnitude of impact of PanK4 on muscle fatty acyl-carnitine levels, which exhibited significant reductions independent of carbon chain length in the absence of PanK4. Whether this decrease can be solely attributed to a defect in the carnitine shuttle, which regulates the transport of long-chain fatty acids from the cytosol into the mitochondria, remains to be elucidated. From a functional standpoint, it is intriguing that the loss of PanK4 leads to a 25% decrease in FAOX in oxidative SkM, which may be linked to the higher levels of malonyl-CoA, an allosteric inhibitor of CPT1. For comparison, deletion of AMPKα, a recognized key regulator of FAOX in SkM, results in a 20% reduction in FAOX in oxidative soleus muscle[35]. A limitation of these findings is that only radiolabeled palmitic acid (C16:0) was studied, leaving it unclear whether other fatty acids with different chain lengths and saturation would also show impaired oxidation.

Our findings suggest that PanK4 is a positive regulator of both FAOX and glucose uptake. Typically, glucose and fatty acid utilization are inversely related, meaning that increased use of one occurs at the expense of the other. Yet, we observed that both SkM glucose uptake and FAOX were compromised, when PanK4 was lacking. This may stem from fiber-type-dependent metabolic effects of PanK4. A limitation of our study is that we did not systematically assess these effects across different muscle fiber types. However, we measured insulin-stimulated glucose uptake in both oxidative soleus and highly glycolytic EDL muscles, and our results suggest that PanK4 primarily regulates glucose uptake in glycolytic fibers. Regarding the in vivo exercise data, we observed a significant effect of PanK4 on general SkM glucose uptake during running, with the largest numerical impairment occurring in quadriceps muscle, which contains a high proportion of glycolytic fibers. Yet, whether there is a fiber-type dependent effect of PanK4 during exercise on glucose uptake requires clarification. We assessed FAOX only in the oxidative soleus muscle, as measuring FAOX in glycolytic EDL muscle is more challenging due to its threefold lower FAOX activity, likely due to reduced mitochondrial content. While we can conclude that FAOX is impaired in soleus muscle lacking PanK4, it remains unclear if this extrapolates to highly glycolytic SkM. Further investigation into fiber-type-specific differences is needed to better understand PanK4's role in muscle metabolism.

Another possibility for PanK4 regulating both glucose and lipid metabolism may be related to the high levels of acetyl-CoA, which is considered an important cellular energy gauge that regulates energy substrate use[24]. Persistently high levels of acetyl-CoA, especially in conditions when energy demands are low, could represent a form of cellular overnutrition that disrupts both glucose uptake and FAOX.

Clearly more work is required to understand whether the elevated acetyl-CoA levels are central to the metabolic impairment observed in the absence of Pank4 in muscle.

An intriguing question remains regarding the source of increased acetyl-CoA: whether it results from an oversupply derived from certain metabolic processes or an underutilization by acetyl-CoA-consuming pathways. Overall, our data indicate that glucose, lipid, and protein catabolism do not contribute to the elevated acetyl-CoA levels. We found preliminary evidence for an interaction between PanK4 and BDH1, an enzyme facilitating the conversion of the ketone body, beta-hydroxybutyrate, into acetyl-CoA. An interesting hypothesis to test will be whether increased uptake of ketone bodies from the circulation contributes to elevated acetyl-CoA production in PanK4 mKO SkM. Another possibility is that acetyl-CoA-consuming pathways such as polyamine metabolism might be altered, which has been suggested to regulate acetyl-CoA content[36]. It will also be key to understand where in the cell acetyl-CoA is increased as there are distinct actions of acetyl-CoA in mitochondria, cytosol, and nucleus[24].

It also remains to be determined what energy substrate SkM is using if both glucose uptake and lipid utilization is compromised. This question could also be related to fiber-type differences, where glycolytic muscle fibers may rely more on FAOX in the face of impaired glucose uptake. Whereas it may be the other way around in more oxidative muscle, where FAOX might be impaired. Alternatively, it may be that SkM together with the rest of the organism adapts to use alternative fuels such as ketone bodies, short-chain fatty acids, or certain amino acids. Notably, we found that glycogen content is increased in SkM lacking PanK4. High glycogen levels may help fuel SkM energy demands during exercise in PanK4 mKO mice and compensate for impaired glucose uptake. Interestingly, it has also been proposed that glycogen levels can modulate insulin-stimulated glucose uptake, with high glycogen levels possibly decreasing insulin-controlled glucose uptake (compared to lower glycogen levels) particular in glycolytic muscles[37]. It is important to highlight that basal or resting glucose uptake was largely normal in PanK4 mKO mice and impairments in glucose uptake were uncovered when challenged with insulin or exercise. Thus, the reliance on an alternative energy substrate might not be particularly relevant in a resting mouse in the postprandial state.

In addition to its role in SkM, it will be important to explore the role of PanK4 in the central nervous system. PanK4 is highly expressed in the hypothalamus and pituitary gland (Fig. 1C). Considering the growth/IGF-1 phenotype of the global PanK4 knockout animals, it is tempting to hypothesize that PanK4 plays a role in regulating the release of growth hormone from the pituitary gland.

Overall, we show that PanK4 is a conserved exercise target with high abundance in SkM, where it regulates acetyl-CoA levels, glucose uptake and fatty acid oxidation.

## Methods

Animal experiments were performed in accordance with the Animal Protection Law of the European Union, and upon permission by the Government of Upper Bavaria, Germany, by the ethics committee of the State Office of Environment, Health and Consumer Protection of Brandenburg, Germany, or by the Danish Animal Experimentation Inspectorate. Samples from two previously published human studies[38,39] were analyzed. Both studies were approved by the regional ethics committee in Denmark (Journal numbers H-1-2012-006 and H-18051389), and written informed consent was obtained from all participants.

### Animals experiments and housing conditions
Mostly male mice were used, as they are more prone to exhibit dysregulation in glycemia than female mice. In a few critical experiments especially related to the alteration in acetyl-CoA and associated metabolites, female mice were included. Mice were group-housed whenever possible. Unless stated otherwise all experiments were performed at ~22 °C with a 12:12 h light-dark cycle. Mice had *ad libitum* access to water and chow diets (Altromin 1324, Brogaarden, Denmark, Altromin 1314, Altromin GmbH, Lage, Germany, or ssniff V1534-300, ssniff Spezialdiäten GmbH, Soest, Germany).

### Methods Related to Figs. 1 and S1
**Human Exercise Study 1.** The detailed experimental procedure can be found elsewhere[38]. Briefly, $n = 6$ self-reported males (mean age 27 years and mean BMI of 24 kg/m²) participated in a single vigorous cycle session (77%-88% Watt_max) until exhaustion, which typically occurred after 8–11 minutes of total exercise time. Biopsies from the vastus lateralis were taken before (PRE), immediately after cessation of exercise (0 h post EX) and 2 and 5 hours post-exercise (2 h and 5 h post EX).

**Human Exercise Study 2.** The detailed experimental procedure can be found elsewhere[39]. Briefly, $n = 8$ self-reported males (26.3 ± 1.3 years and mean BMI of 24 kg/m²) underwent three experimental acute exercise trials in a randomized order, and each trial was separated by at least 10 days. The endurance trial consisted of 90 min of continuous cycling at 60% VO2 peak. The sprint trial consisted of a 5-min warmup at 50 W, followed by three bouts of 30-s all-out sprint (Wingate tests) on an ergometer bike. Each Wingate test was interspersed by 4 min of active recovery at 5 W. The resistance trial was bilateral knee extension, with a warmup consisting of 3 sets of 10 repetitions with a load corresponding to 50% of the 10-RM load, followed by 6 sets of 10 repetitions with 2-min rests. Muscle biopsies were obtained from the vastus lateralis before (Pre EX), immediately after (0 h post EX) and 3 h after exercise cessation (3 h post EX).

**RNA extraction and RT-qPCR.** RNA from tissues was isolated by Trizol-Chloroform (peqGOLD TriFastTM, 30-2010P, VWR) extraction in homogenized tissues and precipitation with Isopropanol. Genomic DNA was removed (DNAse I, EN0521, Fisher Scientific; RiboLock RNAse Inhibitor, EO0382, Fisher Scientific) and 1 µg RNA was transcribed into cDNA using LunaScript RT Super Mix (E3010L, New England Biolabs). mRNA levels were determined by qPCR using SYBR Green (Luna Universal qPCR Master Mix, M3003E, New England Biolabs) on a VIA7 Real-Time PCR System (4453534, Thermo Fischer Scientific). Primer efficiency was determined with an experiment-specific standard curve and target gene expression was normalized to mRNA level of housekeeping genes as indicated in the figure legends. A list of primers used in this manuscript can be found in Supplementary Table 1.

**Whole genome transcriptome analysis.** Gastrocnemius muscles from male PanK4 WT and PanK4 mKO mice injected with glucose 40 min prior to euthanizing them at age 28 weeks were used and RNA was extracted as described above (see "Gene Expression"). The RNA sequencing was carried out by the BGI Genomics laboratory in Hong Kong via nanoball sequencing technology. Raw 150 bp paired reads have been adapter trimmed and quality controlled. The filtered sequences resulted in ~ 9GB reads per sample. Subsequently, additional quality control was assessed on the 9GB filtered reads using FASTQC V0.12.1 and aligned to the mouse reference GRCm39.110 using STAR (V2.7.11a). The gene abundance was calculated by Stringtie (V2.2.1) and expressed by fragment per kilobase of exon per million (FPKM). Differentially gene expression was calculated using DESeq2 (V1.34.0).

**Protein extraction and Western blotting.** Tissues were homogenized in bead-mill homogenizers in ice-cold homogenization buffer (10% Glycerol, 20 mM Na-pyrophosphate, 150 mM NaCl, 50 mM HEPES (pH 7.5), 1% NP-40, 20 mM β-glycerophosphate, 10 mM NaF, 2 mM PMSF,

1 mM EDTA (pH 8.0), 1 mM EGTA (pH 8.0), 10 µg/ml Aprotinin, 10 µg/ml Leupeptin, 2 mM Na3VO4, 3 mM Benzamidine. Lysate supernatant was obtained by centrifugation for 20 min at 13,000 × g at 4 °C. Lysate protein content was determined with the bicinchoninic acid (BCA) method using BSA standards (Pierce) and BCA assay reagents (Pierce) and all lysates were diluted to the same protein concentration with double deionized water. Total protein and phosphorylation levels of indicated proteins were determined by standard immunoblotting technique loading equal amounts of protein. For determining pan-lysine acetylation a modified buffer was used that contained deacety-lase inhibitors: 10% Glycerol, 20 mM Na-pyrophosphate, 150 mM NaCl, 50 mM HEPES (pH 7.5), 1% NP-40, 20 mM β-glycerophosphate, 10 mM NaF, 1 mM EDTA (pH 8.0), 1 mM EGTA (pH 8.0), 10 µg/ml Aprotinin, 10 µg/ml Leupeptin, 2 mM Na3VO4, 3 mM Benzamidine, 1 mM Na-butyrate, and 5 mM Nicotinamide. List of antibodies used can be found in Supplementary Table 2. For Figs. 1F and S1E, F, indicated tissues were collected analyzed from four male 20-week-old C57BL/6L mice.

**Retrieving GWAS data for *PANK4*.** The GWAS data for PANK4 was obtained by downloading information from the NHGRI-EBI GWAS catalog (https://doi.org/10.1093/nar/gkac1010) using FTP servers to examine regions 50 kb up- and downstream of the PANK4 Gene. Sub-sequently, the Manhattan plot was generated using R-version 4.2.3, utilizing a modified version 0.18 of qqman. This adapted version allowed color coding for variant consequences and replacement of chromosomes with phenotypes. Variant consequences were obtained using the biomaRt R-package version 2.54.

**rAAV6-mediated overexpression of PanK4 in skeletal muscle.** rAAV6 generation - Generation of cDNA constructs encoding murine N-terminal Flag-tagged wild-type (WT) PanK4 and a mutant PanK4 in which Ser63 was exchanged with alanine (S63A) were synthesized and subcloned into a recombinant adeno-associated (AAV) expression plasmid consisting of a cytomegalovirus (CMV) promoter/enhancer and SV40 poly-A region flanked by AAV2 terminal repeats (pAAV2) by Genscript (Piscataway, USA) described previously[40,41]. Co-transfection of HEK293T cells with the above-mentioned pAAV2 and the pDGM6 packaging plasmid generated type-6 pseudotyped viral vectors that were harvested and purified as described[40,41] and are herein referred to as rAAV6:PanK4 or rAAV6:PanK4S63A. In short, HEK293T cells were plated at a density of $8 \times 10^6$ cells onto a 15 cm culture dish and 8–16 h later co-transfected with 22.5 µg of a vector genome-containing plas-mid and 45 µg of the packaging/helper plasmid pDGM6 using the calcium phosphate precipitate method. Seventy-two hours after transfection, cells, and culture medium were collected and homo-genized before 0.22 µm clarification (Millipore). The vector was pur-ified from the clarified lysate by affinity chromatography using a HiTrap heparin column (GE Healthcare), ultracentrifuged overnight, and resuspended in sterile Ringer's solution. The purified vector pre-parations were titred with a customized sequence-specific quantitative PCR-based reaction (Applied Biosystems) (Life Technologies).

**Transduction of mouse muscles using rAAV6 vectors.** Three months old male C57BL/6J mice (Janvier) placed under general anaesthesia (2% isofluorane in O2) to enable right TA muscle administration of $1.0 \times 10^9$ vector genomes in a volume of Gelofusine (30 µl) of rAAV6-PanK4 or rAAV6:PanK4S63A with contralateral TA muscles receiving corre-sponding vg of control rAAV6:MCS (multiple cloning site). Mice were euthanized 14 days later; TA muscles were collected and PanK4 and p-PanK4 Ser63 were assessed by Western blotting.

**Generation of *Pank4^-/-* (PanK4 KO) mouse.** To generate a Pank4 knockout (KO) mouse embryonic stem cell (mESC) the open reading frame of *Pank4* was replaced with a previously published cassette NLS-lacZ-2A-H2B-Venus-neo cassette[21]. Homologous arms (HA) were

generated using primers "Pank4 Ex2 fwd" and "PanK4 Ex2 rev" (5´HA, 1919 bp) and "Pank4 Ex19 fwd" and "Pank4 Ex19 rev" (3´HA 1317b bp). The pBKS-NLS-lacZ-2A-H2B-Venus-neo vector was digested with the enzymes, NotI, SacII, SalI, and KpnI. Resulting fragments were purified and used together with both HA in Gibson cloning to generate the "Pank4 KO NLS-lacZ-2A-H2B Venus targeting vector". By annealing two single-stranded (ss)DNA oligos, two gRNAs with target sites in Exon2 and Exon 19 of Pank4 were cloned into the BbsI site of the pbs-U6-chimaric RNA vector. Both gRNA vectors were mixed with an expres-sion vector of Cas9-nickase (pCAG-Cas9v2D10A-bpA; both vectors were generous gifts from O. Ortiz, Institute of Developmental Genet-ics, Helmholtz München, Germany) and together with the targeting vector were used for transfection of IDG3.2 F1-Hybrid mES cells[42]. After selection with 300 µg /ml G418, clones were analyzed for correct integration of the targeting vector by PCR genotyping and sequencing (primers #1, 2, 3, and 4). Positively targeted mESs were aggregated with CD1 morulae to obtain mouse chimeras. The offspring of the chimeras were genotyped to test for germline transmission of the targeted allele (using primers #5, 6, and 7): 790 bp and 559 bp for the wild type or KI allele, respectively. The floxed Neo cassette was deleted by crossing mice with C57BL/6N-*Gt(ROSA)26Sor^tm16(cre)Arte* mice and progeny was genotyped for the absence of the selection cassette (using primers #8, 9 & 10): 317 bp and 459 bp for Neo allele product and delta-Neo allele product, respectively. Mice were backcrossed to C57BL/6 J back-ground for at least six generations. Three PanK4 WT and three PanK4 KO mice were euthanised at age 12 weeks and PanK4 and p-PanK4 Ser63 were assessed by Western blotting in TA and gastrocnemius muscles. gRNA and primer sequences used for generating and geno-typing PanK4 KO mice can be found in Table 1.

### Methods related to Figs. 2 and S2–4

**PanK4 KO phenotyping.** The generation of PanK4 KO mouse is described above. Male and female PanK4 WT and KO mice were kept on standard chow diet and body weight was determined weekly from age 5 weeks until age 20 weeks. Mice were killed by isoflurane over-dose, body length was measured and blood and tissues were collected.

**Plasma IGF-1.** IGF-1 in plasma was determined using an ELISA from R&D systems (Mouse/Rat IGF-I/IGF-1 ELISA Kit – Quantikine; Cat# MG100).

**Generation of *Pank4^flox/flox* and muscle-specific *Pank4* KO (*Pank4^flox/flox*:*HSA-Cre*) mouse.** The floxed *Pank4* mouse line was created by flanking exon 2 to 9 with loxP sites. Targeting vector constructs were designed as 5´HA(4.3 KB)-loxP-Pank4(exon2-9)-FRT-neomycin-FRT-loxP-3´HA (4.2 KB) using the primers from Table 2. The targeting vector was injected into embryonic stem cells from C57BL/6J background mice, and a Neo-resistant strain was estab-lished. Floxed *PanK4* mice were obtained from intercrosses with *CAG-FLP* transgenic mice (C57BL/6N-*Gt(ROSA)26Sor^tm1(Flp1)Dym*. *HSA–Cre* (stock #006149) transgenic mice were obtained from Jackson Laboratory (Bar Harbor, ME). Homozygous floxed *Pank4* mice were then crossed with *HSA–Cre* mice to obtain muscle-specific Pank4 KO (*Pank4^flox/flox*:*HSA-Cre^-/+*) mice which were verified by PCR, Western blotting and qPCR. Floxed PanK4 mice only (*Pank4^flox/flox*:*HSA-Cre^-/-*) were used as wild-type (WT) mice in the study. LoxP and HSA-Cre genotyping were done with the specific primers listed in Table 3.

**PanK4 mKO phenotyping.** Male PanK4 WT and PanK4 mKO were fed standard chow diet and body weight and food intake were determined weekly. Glucose tolerance was determined at age 12 and 27 weeks. Body composition was assessed at age 26. At age 28 weeks mice were fasted for 5 h before being i.p. injected with glucose (2 g glucose per kg body weight) and killed 40 min later.

**Table 1 | Primers/gRNA used for generating/genotyping PanK4 KO mice**

| Oligo name | Sequence (5′->3′) |
|---|---|
| PanK4 Ex2 | GACTCACTATAGGGCGAATTGGAGCTCCACCGC TGTGCGCCACCACTCCTGATG |
| PanK4 Ex2 | TTGTTTTTCGAGCTTCAAGGTTCATGGTGGCGCATCCACCTGCACGGACAAGACAAAAAC |
| PanK4 Ex19 | GGGCTGCAGGAATTCGATAGCTTGGCTGCAG GTGAGATACCCACTCTGCCAGACG |
| PanK4 Ex19 | AACCCTCACTAAAGGGAACAAAAGCTGGGTA CTGCCTGCCTGGAATTTGAGAAAGAC |
| Pank4 gRNA #1.1 | CACCGGGGGGTGGAATAGTATGCCAACT |
| Pank4 gRNA #1.2 | AAACAGTTGGCATACTATTCCACCCCC |
| Pank4 gRNA #2.1 | CACCGGGAGTGAGATCTTTTGACCACC |
| Pank4 gRNA #2.2 | AAACGGTGGTCAAAAGATCTCACTCCC |
| PanK4 #1 | TTGGCAGTGTGCAGTATTCC |
| PanK4 #2 | TTTCGAGCTTCAAGGTTCATGG |
| PanK4 #3 | GGATACCCTAGGCTCGCATAAC |
| PanK4 #4 | GGCTGGACGTAAACTCCTCTTC |
| PanK4 #5 | GAGTGCTCTCCAAGGCAAAC |
| PanK4 #6 | CTGCAGTGGCTTGCTGAATC |
| PanK4 #7 | TAAGAGGCCTGCGTACCTTC |
| PanK4 #8 | AGCCATACCACATTTGTAGAGG |
| PanK4 #9 | CAGCCCAATTCCGATCATATTC |
| PanK4 #10 | ATTGCATCGCATTGTCTGAGTAG |

**Table 2 | Primers for Floxed PanK4 targeting vector**

| Name | Sequence 5′->3′ | Restriction Enzyme Inserted | Product size |
|---|---|---|---|
| Pank4-A1-F | CGATGGTACCGTATACCAAGTAGGCTGGAGTAGC | KpnI/Bstz17I | 462 bp |
| Pank4-A1-R | CGATCTCGAGCTCAAGAGGGAGTCAGATCTCATTACAG | XhoI/SacI | |
| Pank4-A2-F | CGATATCGATACTGGTGTGTCTGAAGTCAGC | ClaI | 4029 bp |
| Pank4-A2-R | CGATGCGATCGCACATGAGGAATGGATCTAGAACATTC | SgfI | |
| Pank4-B-F | CGATGGATCCGATATCCTCCATGAGGATGAAATCCACTGTGGC | BamHI/EcoRV | 384 bp |
| Pank4-B-R | CGATGTATACCAAGGGCTGCTGGGTCG | Bstz17I | |
| Pank4-C1-F | GCTGGTACCGGCGCGCCTCGAGGTCCTCATCTGCTCAGAGTCC | XhoI | 386 bp |
| Pank4-C1-R | CTTGGAGTAGGGATATCCTGGATCTACGAATAGAGAGACC | EcoRV | |
| Pank4-C2-F | GTAGATCCAGGATATCCCTACTCCAAGGTGAGC | EcoRV | 473 bp |
| Pank4-C2-R | TCCTCTTCAGACCTGGCGGCCGCGCCTCAACGAGGACTGAAAACA | NotI | |

**Body composition.** Body composition was analyzed in awake mice using a magnetic resonance whole-body composition analyzer (EchoMRI, Houston, TX).

**Indirect calorimetry.** Energy expenditure, substrate utilization (respiratory exchange ratio, RER), and home-cage activity were measured in single-housed mice using a climate-controlled indirect calorimetry system (TSE System, Bad Homburg, Germany). Indirect calorimetry was conducted on chow-fed 14-18-week-old male PanK4 WT and PanK4 mKO mice. For the first 144 hours, the mice were fed standard chow diet, with a 12-hour overnight fasting period during which food was removed between 60 and 72 hours. After 144 hours, mice received high-fat diet with 60 kcal% fat (Research Diets D12492) for 48 hours.

**Nontargeted metabolomics.** *Muscle Tissue Collection and Preparation for Metabolomics* - Tibialis anterior (TA) and gastrocnemius muscles from male PanK4 WT and PanK4 mKO mice injected with glucose 40 min prior to euthanizing them at age 28 weeks were used. Muscles were stored at −80 °C and were randomized prior sample extraction. Muscle pieces (~30–100 mg) were first homogenized with 1.4 mm ceramic beads in water (15 μl/mg tissue) at 10 °C. To extract metabolites and to precipitate the protein, 500 μL methanol extraction solvent containing recovery standard compounds was added to each 100 μL of

tissue homogenate. Supernatants were then aliquoted. Two (i.e., early and late eluting compounds) aliquots were dedicated for analysis by UPLC-MS/MS in electrospray positive ionization and one for analysis by UPLC-MS/MS in negative ionization. The extract aliquots were dried under nitrogen stream (TurboVap 96, Zymark) and stored at −80 °C until the UPLC-MS/MS measurements were performed. Three types of quality control samples were included in each plate: samples generated from a pool of human ethylenediamine tetraacetic acid (EDTA) plasma, pooled samples generated from a small portion of each experimental sample served as technical replicate throughout the data set, and extracted water samples served as process blanks.

*UPLC-MS/MS Non-targeted Measurements* - The UPLC-MS/MS platform utilized a Waters Acquity UPLC with Waters UPLC BEH C18-2.1 × 100 mm, 1.7 μm columns, a Thermo Scientific Q Exactive high resolution/accurate mass spectrometer interfaced with a heated electrospray ionization (HESI-II) source, and an Orbitrap mass analyzer operated at 35,000 mass resolution.

For acidic positive electrospray ionization conditions, that chromatographically optimized for more hydrophilic compounds (for early eluting compounds), the extracts were gradient eluted from the C18 column (2.1 × 100 mm Waters BEH C18 1.7 μm particle) using water and methanol containing 0.05% perfluoropentanoic acid (PFPA) and 0.1% formic acid. Another extract aliquot that was also analyzed using acidic positive electrospray ionization conditions, but was

**Table 3 | Primers for Floxed PanK4 and HSA-Cre genotyping**

| Name | Sequence 5´->3´ | Product size |
|---|---|---|
| Pank4-A1 Loxp-F | GCCTTGAACCCAGATTCGTTAAGAACAG | WT:253 bp |
| Pank4-A2 Loxp-R | GATGAGAGGAGAGAGTACCGATCGC | Mut:315 bp |
| Pank4-Frt-F | CCTGGAATGTTCTAGATCCATTCCTC | WT:269 bp |
| Pank4-Frt-R | CCAGACTTATATGGTGTCTGAGCTAC | Mut:399 bp |
| Flp-F | TCAGCGATATTAAGAACGTTGATCCG | Mut: 224 bp |
| Flp-R | TGAAGAATTGCCGGTCCTATTTACTCG | |
| HSA-Cre-F | CGCATATGCTTCCATCCCCC | WT: - |
| HSA-Cre-R | AAGTAGAGTGGGGGTCAGCA | Cre: 350 bp |
| Pank4 sense | TAAGGAGCCTCCCAACCCACCA | WT: 124 bp |
| Pank4 wt_anti | GAATGTTCTAGATCCATTCCTCATGTCTC | Floxed: - |
| Pank4 sense | TAAGGAGCCTCCCAACCCACCA | WT: - |
| Pank4 flox_anti | GCTATACGAAGTTATTAGGTGGATCCGA | Floxed: 130 bp |
| Cre sense | ACGGACAGAAGCATTTTCCAGGT | WT: - |
| Cre anti | CGGTCGATGCAACGAGTGATG | Cre: 100 bp |

chromatographically optimized for more hydrophobic compounds (for later eluting compounds), was gradient eluted from the same C18 column using methanol, acetonitrile, and water; containing 0.05% PFPA and 0.01% formic acid and was operated at an overall higher organic content. The basic negative ion condition extracts were gradient eluted from a separate C18 column using water and methanol containing 6.5 mM ammonium bicarbonate at pH 8.

The MS analysis alternated between MS and data-dependent MS2 scans using dynamic exclusion and a scan range of 80–1000 m/z. Metabolites were identified by automated comparison of the ion features in the experimental samples to a reference library of chemical standard entries that included retention time, molecular weight (m/z), preferred adducts, and in-source fragments as well as associated MS spectra and curation by visual inspection for quality control using proprietary software developed by Metabolon Inc. Only fully annotated metabolites were included for further evaluation. Data were normalized according to raw area counts, and then each metabolite scaled by setting the median equal to 1. Missing data were imputed with the minimum. Biochemicals labeled with an asterisk (*) indicate compounds that have not been officially confirmed based on a standard, but we are confident in its identity.

*Data Processing and Analyses* - Annotations of different experiments have been merged and Welch's *t*-test was conducted, comparing two groups to calculate paired or unpaired raw p values. Calculations have been performed using R Version 4. Volcano plots and heatmaps have been created using GraphPad Prism.

**Fatty acid oxidation ex vivo.** Exogenous palmitate oxidation in isolated soleus muscles of PanK4 and mKO mice was analyzed as previously described[43]. Briefly, the soleus muscles were taken from anesthetized male mice fasted for 3 hours aged 12–19 weeks, and mounted on a force transducer in a 15 ml container, where they bath at resting tension of 4–5 mN in a Krebs-Henseleit bicarbonate buffer (pH 7.4, 5 mM glucose, 2% FA-free bovine serum albumin, 0.5 mM palmitic acid) at 30 °C for 20 minutes. During either rest or contraction (30 Hz, 600 ms pulse duration, 18 tetani/min for 25 minutes), the buffer was exchanged for a fresh buffer that also contained palmitate [1–14 C] palmitate (0.0044 MBq/ml; Amersham BioSciences, Buckinghamshire, United Kingdom). To seal the incubation chambers, mineral oil (Cat. No. M5904, Sigma–Aldrich) was added on top. At the end of the 25 min-long rest or contraction (18 trains/min, 0.6 s pulses, 30 Hz, 60 V) protocol, incubation buffer was collected and the muscles were flash-frozen in liquid nitrogen to determine the rate of palmitate oxidation as described[43]. Palmitate oxidation was determined as $CO_2$

production (complete FA oxidation) and acid-soluble metabolites (representing incomplete FA oxidation). As no difference was observed in complete and incomplete FA oxidation between genotypes, palmitate oxidation is presented as a sum of these two forms.

**Citrate synthase activity.** The activity of Citrate Synthase (CS) was analyzed spectrophotometrically as previously described[44] by tracking the generation of DTNB at 412 nm. The process involved homogenizing soleus muscle in a solution of 50 mM Tris, 1 mM EDTA (pH 7.4), and 0.1% Triton X-100, and then centrifuging the mixture for 10 minutes at 4 °C. The supernatant was utilized to measure protein content and levels of CS activity. 10 μL of a 1:6 diluted tissue sample was placed in one well of a 96-well plate. Afterwards, 215 μL of reaction buffer (100 mM Tris, 1 mM MgCl2, 1 mM EDTA (pH 8.2), and 0.1 M DTNB) and 25 μL of Acetyl CoA (3.6 mM) were added. The procedure was repeated in triplicate. The reaction was initiated by adding 50 μL of Oxaloacetate (3 mM) and the change in absorbance at 412 nm was monitored for 10 minutes at 37 °C. The CS activity was calculated based on the slope of the linear part and normalized to mg of protein.

**Tissue triglycerides.** Tissue triglycerides (TG) were determined as previously described[45] using the kit RandoxTR-210 (Randox) using 20–30 mg of the indicated muscle that was pulverized in liquid nitrogen.

**Plasma fatty acids (FA).** The concentrations of plasma FA (NEFA C kit; Wako Chemicals, Denmark) were measured colorimetrically on an autoanalyzer [Pentra C400 analyzer (Horiba, Japan)].

**Other.** Protein and RNA extraction as well as Western blotting and qPCR were conducted as described above and antibodies and primer sequences are listed in supplementary tables 1 and 2

**Methods related to Figs. 3 and S5.** Nontargeted Metabolomics has been described above.

**Targeted metabolite analysis.** For the fasted versus fed experiment, male and female PanK4 WT and PanK4 mKO mice at age 12-17 weeks (age was balanced across groups) were overnight fasted for 14 hours. Afterwards some mice had *ad libitum* access to a chow diet for two hours while some mice remained without food during that time. Mice were then euthanized by cervical dislocation and skeletal muscles were quickly resected out. For analysis of malonyl-CoA in fed mice (Fig. 3G), male PanK4 WT and PanK4 mKO mice at age 15–25 weeks (age was balanced across groups) were euthanized by cervical dislocation after a one-hour fast two hours into the light cycle and skeletal muscle were quickly resected out.

Approximately 20 mg of mouse gastrocnemius were extracted as follows: 360 μL of 80% methanol containing 0.1 M formic acid and internal standards (listed in supplementary tables 3 and 4) were added to the samples on ice, immediately after the addition of extraction solvent the samples were vortexed for 5 seconds and 20 μL of 9% ammonium bicarbonate (w/v in water) were added; the samples were vortexed again for 5 seconds and each sample was shaken with a tungsten carbide bead at 30 Hz for 3 minutes, then the samples were centrifuged at 14,000 g for 10 minutes. The supernatant was transferred to LC vials and dried using a centrifugal concentrator (Thermo Fisher Scientific, Waltham, USA), and finally reconstituted in 50 μL of 50% methanol. Subsequently, the samples were analyzed using a liquid chromatography-tandem mass spectrometry (LC-MS/MS) system consisting of an Agilent 1290 UPLC connected to an Agilent 6490 triple quadrupole (Agilent, CA, USA), metabolites were detected in multiple-reaction monitoring (MRM) mode.

Analysis of malonyl-CoA in gastrocnemius and quadriceps muscles was conducted as follows: first, 10 μL of 50 μM malonyl-CoA-$^{13}C_3$

and 10 μL of 50 mM ascorbic acid were added to the sample, then 360 μL of 0.1 M formic acid in 80% aqueous methanol was added and the sample was vortexed for 5 s, the sample was then neutralized by adding of 9% ammonium bicarbonate in water. Each sample was then shaken with a tungsten carbide bead at 30 kHz for 3 minutes, then the samples were centrifuged at 14,000 x g for 10 minutes and the supernatant was transferred to an LC vial and evaporated by using a centrifugal concentrator (Thermo Fisher Scientific, Waltham, USA). Prior to the analysis, each sample was reconstituted in 50 μL of aqueous methanol, then 3 μL were injected into column (iHILIC-(P) Classic, 30×2.1, HILICON AB, Sweden) and separated at a flow rate of 0.6 ml/min with the following gradient: 0.0 min (85% B), 0.5 min (85% B), 3 min (70% B), 3.9 min (70% B), 4 min (50% B), 4.5 min (50% B), 4.6 min (85% B), 6.0 (85% B). Malonyl-CoA was detected in the MRM mode by Agilent 6495D triple quadrupole (Agilent, CA, USA). All other ion source parameters and composition of the mobile phase were identical with the general method for analysis of phosphorylated metabolites that is described below.

The separation and quantification of phosphate-containing metabolites (Table S4, Fig. S4A, B) were achieved by injecting 3 μL of the extracts to a HILIC column (iHILIC-(P) Classic, PEEK, 50 × 2.1 mm, 5 μm, HILICON, Umeå, Sweden), the column and autosampler were maintained at 40 °C and 4 °C, respectively. The mobile phase is composed of (A) 10 mM ammonium acetate with 5 μM medronic acid in water of pH 6.8 and (B) 10 mM ammonium acetate in 90% acetonitrile. The mobile phase was delivered at a flow rate of 0.35 ml/min at the following gradient elution: 0.0 min (85% B), 5 min (60% B), 7 min (30% B), 8 min (30% B), 9 min (85% B), 15 min (85% B). Analytes were ionized in an electrospray source operated in the positive mode. The source and gas parameters were set as follows: ion spray voltage 4.0 kV, gas temperature 200 °C, drying gas flow 11 l/min, nebulizer pressure 30 psi, sheath gas temperature 375 °C, sheath gas flow 12 l/min, fragmentor 380 V. Quantification of AMP, ADP, ATP, Mal-CoA, and Ac-CoA was conducted based on calibration curve using their isotopically labeled analogs as internal standards. External calibration curves were used for quantification of Suc-CoA, CoA, and 3-dp-CoA because of the unavailability of their isotopically labeled standards. The calibration for all phosphate-containing metabolites covered linear dynamic range from 5 nM to 50 μM.

Carboxylic acids were separated and quantified by using LC-MS/MS after their derivatization with 3-nitrophenylhydrazine (3-NPH). The samples were derivatized according to a published method[46] with modifications, briefly: 20 μL of 120 mM EDC (dissolved in 6% pyridine in 50% methanol) and 20 μL of 200 mM 3-NPH (dissolved in 50% methanol) were consecutively added to 20 μL of an extract. The sample was incubated at room temperature (21 °C) for 60 minutes, afterwards 40 μL of 0.05 mg/ml BHT (dissolved in pure methanol) were added to the samples and vortexed. Eventually, 5 μL of each sample were analyzed on an Acquity UPLC HSS-T3 column (100 × 2.1 mm, 1.8 μm, Waters, MA, USA) by using a gradient elution of 0.1% formic acid (v/v) in water as mobile phase A and 0.1% formic acid in acetonitrile as mobile phase B – resulting in separation. The mobile phase was delivered on the column by a flow rate of 0.35 ml/min with the following gradient: 0 min (5% B), 12 min (100% B), 13 min (100% B), 14 min (5% B), 17 min (5% B). Column and autosampler were thermostated at 30 °C and 4 °C, respectively. Analytes were ionized in an electrospray ion source operated in the negative mode. The source and gas parameters were set as follows: ion spray voltage −3.5 kV, gas temperature 150 °C, drying gas flow 11 l/min, nebulizer pressure 30 psi, sheath gas temperature 400 °C, sheath gas flow 12 l/min, fragmentor 380 V. The instrument was operated in dynamic multiple reaction monitoring mode (MRM), and the MRM transitions (Table S2) of derivatized carboxylic acids were optimized by using Agilent MassHunter Optimizer. Malate, citrate, pyruvate, 2-oxoglutarate, fumarate, succinate, lactate, and 2-hydroxyglutarate were quantified based on internal calibration

with their isotopically labeled standards. Analysis of acids lacking their isotopically labeled analogues (oxaloacetate, isocitrate, cis-aconitate, itaconate) was conducted through external calibration curve. The calibration for all carboxylic acids covered a linear dynamic range from 80 nM to 60 μM.

**Pyruvate Dehydrogenase alpha (PDHα) activity.** PDHa activity was measured after homogenizing 10–15 mg of wet weight muscle tissue and snap-freezing the homogenate in liquid nitrogen as described previously[47] and PDHa activity was normalized to creatine concentration in each muscle sample to correct for non-muscle tissue.

**Other.** Protein and RNA extraction as well as Western blotting and qPCR were conducted as described above and antibodies and primer sequences are listed in supplementary tables 1 and 2.

### Methods related to Figs. 4 and S6–10
**Glucose tolerance test.** Mice were fasted for six hours from 8 a.m. and intraperitoneally injected with 2 g/kg bodyweight D-glucose (0.2 g in 1 ml saline). Blood glucose concentration in mixed tail blood was measured with a glucometer just before glucose injection (0 min) and 15, 30, 60, and 120 min after glucose injection. At the 0 min time-point extra blood was collected for determination of plasma insulin levels.

**Insulin.** Plasma insulin levels were analyzed using the "Mouse Ultra-sensitive Insulin ELISA" from Alpco (80-INSMSU-E01).

**Ex vivo insulin-stimulated 2-deoxyglucose (2-DG) uptake.** Soleus and EDL muscles from both legs were dissected out from anesthetized mice (6 mg pentobarbital and 0.24 mg lidocaine/100 g body wt). Mice were euthanized by cervical dislocation after muscles had been removed. Muscles were gently lengthened to resting tension (4–5 mN) in incubation chambers (Multi Myograph system; Danish Myo-Technology, Denmark). These chambers contained 4 ml heated (30 °C) Krebs–Ringer–Henseleit (KRH) buffer supplemented with 2 mM pyruvate, and 8 mM mannitol. Muscles were incubated for 30 min without or with insulin (Insuman Rapid, Sanofi Aventis) at a concentration of 1.8 nM and 3 nM for soleus and EDL, respectively. During the last 10 min of insulin stimulation, 2-DG uptake was measured with 3H-2-DG and 14C-Mannitol radioactive tracers and 1 mM of 2-DG. Muscles were washed in ice-cold KRH buffer, blotted dry, and snap-frozen in liquid nitrogen, trimmed, and weighed, before being stored at −80 °C.

**Treadmill running experiment.** *Maximal running capacity* – Mice were acclimated to the treadmill on three separate days, running for 10 minutes at 0.17 m/s on a 0° incline. The maximal running capacity test was conducted three days after the last acclimation session. This test was performed on a 10° incline, beginning with a 5-minute warm-up at 0.17 m/s, followed by an increase in speed by 0.02 m/s every minute until exhaustion. The testing was conducted blind to the genotypes.

*Endurance running capacity* – Mice ran at an incline of 10° at a speed of 0.16 m/s for 10 min, followed by 40% of maximal running speed for 20 min and 60% of maximal running speed for 20 min. This was followed by an incremental increase of 0.02 m/s every 2 min until exhaustion. The testing was conducted blind to the genotypes.

*Running-induced glucose uptake* - Blood glucose concentration was measured from mixed tail blood. Subsequently, saline containing 60 μCi/ml 2-[3H]-deoxyglucose (3H-2-DG; Perkin Elmer, Waltham, MA, USA) and 0.1 mM unlabeled 2-deoxyglucose (2-DG) was injected intraperitoneally at 8 μL per gram of body weight. Mice were divided into two groups: one group ran for 20 minutes at a 10° incline and 75% of the overall average maximal running speed, while the other group rested in their cages for the same duration. After 20 minutes, blood glucose concentration was measured again, and additional blood was

collected into heparinized capillary tubes. The mice were then euthanized by cervical dislocation, and their quadriceps, gastrocnemius, and triceps muscles were quickly resected and snap-frozen in liquid nitrogen. Blood samples were centrifuged to obtain plasma, which was also snap-frozen in liquid nitrogen. A 5 μL volume of plasma was used to measure plasma 3H activity via scintillation counting. The area under the curve for 3H-2-DG from 0 to 20 minutes was calculated to estimate circulating 3H-2-DG. Approximately 40 mg portions of quadriceps, gastrocnemius, and triceps muscles were used to determine the clearance of phosphorylated 3H-2-DG (3H-2-DG-6-P) from the plasma into the muscle, as described in Ref. 48. Glucose uptake was estimated by multiplying clearance by the average blood glucose levels.

**Muscle p-PanK4 Ser63 during running in wildtype and kinase-dead AMPK alpha2 (AMPK-KD) mice.** Lysates from gastrocnemius collected after running as described in Ref. 49 were used.

**Ex vivo muscle contraction.** Ex vivo contraction of isolated soleus and EDL muscles were performed as previously described[50]. In short, soleus and EDL muscles were isolated and suspended in incubation chambers with Krebs Ringer Henseleit (KRH) buffer as described above. Muscles were incubated 10 min after which platinum electrodes were placed centrally and on both sides of the muscle to induce contraction. The contraction protocol consisted of 1.0-s trains (100 Hz, 0.2 ms, ~30-40 V) repeated every 15 s for 10 min. After the contraction period, muscles were washed in ice-cold KRH buffer, blotted dry, and snap-frozen in liquid nitrogen before stored at −80 °C.

**In situ muscle contraction.** Rapamycin (1.5 mg/kg) or vehicle control (equal volume of DMSO) was injected intraperitoneally 1 h before muscle contraction. The common tibial nerve of both legs was exposed in anesthetized mice (8 mg pentobarbital and 0.32 mg lidocaine/100 g body wt) after which an electrode was placed on a single nerve followed by in situ contraction of soleus muscle. The contralateral leg served as a sham-operated control. The contraction protocol consisted of 0.5-s trains (100 Hz, 0.1 ms, 5 V) repeated every 1.5 s for 10 min. After the contraction period, soleus muscles were dissected and snap-frozen in liquid nitrogen before stored at −80 °C. Mice were humanely killed by cervical dislocation after the muscles had been removed.

**Ex vivo muscle stretching or AICAR stimulation in wildtype and kinase-dead AMPK alpha2 (AMPK-KD) mice.** Soleus and EDL muscles from wildtype and AMPK-KD were incubated and stimulated with passive stretch (50 nM for 15 min) or with AICAR (2 mM for 40 min) as described in Ref. 51.

**Ex vivo muscle contraction with blebbistatin.** EDL muscle were contracted as described above but only for two min and the contraction force was recorded. Before start on contraction muscles were pretreated with blebbistatin (50 μM) or equal volume of DMSO for 20 min.

**PanK4 overexpression and skeletal muscle glucose uptake and metabolomics.** Generation of vectors and general administration principle can be found above.

*Dose Response* - Three months old male and female C57BL/6J mice (Janvier) were placed under general anaesthesia (2% isoflurane in $O_2$) to enable right TA muscle administration of $1.0 \times 10^9$, $2.5 \times 10^9$, or $5.0 \times 10^9$ vector genomes in a volume of Gelofusine (30 μL) of rAAV6-PanK4 with contralateral muscles receiving corresponding vg of control rAAV6:MCS. Mice were killed 14 days later and PanK4 and p-PanK4[Ser63] abundance in indicated muscles were assessed by WB.

*Glucose Uptake* - Three months old male mice C57BL/6J mice (Janvier) were i.m. injected into right TA with $1.0 \times 10^9$ vector genomes of rAAV6-PanK4 with contralateral muscles receiving corresponding

vector genomes of rAAV6:MCS. 14 days later glucose uptake into TA muscles was assessed in vivo. For this, mice were fasted for 2 h, and mixed tail blood was collected, before mice were IP injected (10 ml/kg body mass) with saline containing 0.1 mM 2-deoxyglucose (2-DG) and 60 μCi/ml 3H-labeled-2-DG[48]. Blood was drawn at 20 min post-injection and mice were euthanized by cervical dislocation and TA muscles were rapidly excised and snap-frozen in liquid nitrogen. Plasma 3H activity was determined by scintillation counting and the area under the curve from 0 to 20 min was calculated by the trapezoid method to estimate the 3H-2-DG levels that the muscles had been exposed to over 20 min [30]. ~30 milligrams of TA muscle was used to determine the accumulation of phosphorylated 3H-2-DG (3H-2-DG-6-P) with the precipitation method[48].

*Nontargeted Metabolomics* - Three months old male C57BL/6J mice (Janvier) were injected intramuscularly into the right TA with $1.0 \times 10^9$ vector genomes of rAAV6-PanK4, while the contralateral muscles received the same amount of rAAV6:MCS. Fourteen days later, some mice were fasted for 14 hours, while others were fasted for 12 hours and then refed for 2 hours with free access to chow. Mice were then euthanized, and TA muscles were quickly resected, frozen in liquid nitrogen, and stored at −80 °C.

Muscle samples (~40 mg) were homogenized with 1.4 mm ceramic beads in water (15 μL /mg tissue) at 10 °C. To extract metabolites and precipitate proteins, 500 μL of methanol extraction solvent containing recovery standards was added to each 100 μL of tissue homogenate. The supernatants were aliquoted, with one aliquot analyzed by UPLC-MS/MS in positive ionization and another in negative ionization. The UPLC-MS/MS platform included a Waters Acquity UPLC with Waters UPLC BEH C18 columns (2.1 × 100 mm, 1.7 μm), a Thermo Scientific Q Exactive mass spectrometer with a heated electrospray ionization (HESI-II) source, and an Orbitrap mass analyzer operated at 35,000 mass resolution.

Two separate C18 columns were used for acidic (solvent A: 0.1% formic acid in water; solvent B: 0.1% formic acid in methanol) and basic (solvent A: 6.5 mM ammonium bicarbonate pH 8.0; solvent B: 6.5 mM ammonium bicarbonate in 95% methanol) mobile phase conditions, optimized for positive and negative ionization respectively. The sample extracts were injected, and the columns developed in a gradient of 99.5% A to 98% B over 11 minutes at a flow rate of 350 μl/min. MS analysis alternated between MS and data-dependent MS2 scans using dynamic exclusion and a scan range of 80–1000 m/z. Metabolites were identified by comparing ion features in experimental samples to a reference library of chemical standards, including retention time, molecular weight (m/z), preferred adducts, in-source fragments, and associated MS spectra, with quality control curation by Metabolon Inc. Only fully annotated metabolites were included for further analysis.

Data were normalized to raw area counts, with each metabolite scaled to a median of 1. Missing data were imputed with the minimum value. Compounds not officially confirmed but confidently identified are marked with an asterisk (*). Annotations from different experiments were merged, and Welch's t-test was conducted to compare two groups, calculating paired or unpaired raw p-values. Calculations were performed using R Version 4. Volcano plots and heatmaps were created using GraphPad Prism.

**PanK4 Interacting partners.** PanK4 was immunoprecipitated from rAAV6-MSC injected TA muscle lysates (400 μg; $n = 2$) with an anti-PanK4 antibody by protein G covalent conjugated agarose beads (Millipore, no. 16-266) or from rAAV6-PanK4 injected TA muscle lysates (400 μg; $n = 4$) with anti-flag coated agarose beads (Millipore, no. A2220) and incubated overnight at 4 °C. As a negative control, anti-PanK4- and anti-flag coated agarose beads-mediated pulldown was also performed in TA muscle lysates (400 ug) from PanK4 KO mice ($n = 3$). The following day, agarose beads were centrifuged, and supernatant was collected. Beads were washed twice with buffer

containing 120 mM HEPES and 240 mM NaCl (pH 7.4) and proteins were eluted with laemmli buffer (1% SDS). Proteins were proteolysed with LysC and trypsin with filter-aided sample preparation procedure (FASP) as described[52,53]. Acidified eluted peptides were analyzed on a QExactive HF mass spectrometer (Thermo Fisher Scientific) online coupled to a Ultimate 3000 RSLC nano-HPLC (Dionex). Samples were automatically injected and loaded onto the C18 trap cartridge and after 5 min eluted and separated on the C18 analytical column (Acquity UPLC M-Class HSS T3 Column, 1.8 μm, 75 μm x 250 mm; Waters) by a 95 min non-linear acetonitrile gradient at a flow rate of 250 nl/min. MS spectra were recorded at a resolution of 60000 with an automatic gain control (AGC) target of 3e6 and a maximum injection time of 30 ms from 300 to 1500 m/z. From the MS scan, the 10 most abundant peptide ions were selected for fragmentation via HCD with a normalized collision energy of 28, an isolation window of 1.6 m/z, and a dynamic exclusion of 30 s. MS/MS spectra were recorded at a resolution of 15000 with a AGC target of 1e5 and a maximum injection time of 50 ms. Unassigned charges, and charges of +1 and >8 were excluded from precursor selection.

### Data processing protocol

Raw spectra were imported into Progenesis QI software (version 4.1). After feature alignment and normalization, spectra were exported as Mascot Generic files and searched against the mouse SwissProt database (16,871 sequences) with Mascot (Matrix Science, version 2.6.2) with the following search parameters: 10 ppm peptide mass tolerance and 20 mmu fragment mass tolerance, one missed cleavage allowed, carbamidomethylation was set as fixed modification, methionine oxidation, and tyrosine, serine, and threonine phosphorylation were allowed as variable modifications. A Mascot-integrated decoy database search calculated an average false discovery of <1% when searches were performed with a mascot percolator score cut-off of 13 and an appropriate significance threshold p.

Peptide assignments were re-imported into the Progenesis QI software and the abundances of all unique peptides allocated to each protein were summed up. The resulting normalized abundances of the individual proteins were used for the calculation of fold-changes of protein ratios between conditions. Statistical analysis was performed on log2 transformed normalized abundance values using Student's t-test. Changes in protein expression between conditions were considered significant at $p < 0.05$.

### Other

Protein, western blotting, whole genome RNA-sequencing and targeted metabolite analyses have been described above. Antibodies are listed in Supplementary Table 2.

### Statistics & reproducibility

Data are presented as means ± SEM in some xy-plots, as individual values only, or as means plus individual data points in bar graphs. No statistical method was used to predetermine the sample size. When appropriate, the ROUT method was used to detect outliers. The experiments were randomized. Whenever possible all or some investigators were blinded to allocation during experiments and outcome assessment. The statistical analyses of data are described in the figure legends. Graphs were prepared and analyzed with GraphPad Prism (version 10).

### Reporting summary

Further information on research design is available in the Nature Portfolio Reporting Summary linked to this article.

## Data availability

Source data are provided with this paper. The raw data of the whole-genome RNA sequencing data generated in this study have been deposited in the GEO database under accession code GSE273694. The mass spectrometry proteomics data have been deposited to the ProteomeXchange Consortium via the PRIDE[54] partner repository with the dataset identifier PXD057017. Further datasets used and/or analyzed during the current study are available as supplementary files. Source data are provided with this paper.

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

## Acknowledgements

The authors would like to acknowledge the critical input of J. F. P. Wojtaszewski (University of Copenhagen) and the assistance with AAV vector design and production of Dr J.R. Davey, Dr. K.I. Watt and Dr H. Qian (University of Melbourne). A.M.F. was supported by a postdoctoral research grant from the Danish Diabetes Academy and by the Novo Nordisk Foundation (grants NNF17SA0031406 and NNF22OC0074110). S.H.R. is funded by the Independent Research Fund Denmark (grant 2030-00007 A) and the Lundbeck Foundation (grant R380-2021-1451). L.L.V.M. is supported by the Lundbeck Foundation (grant R322-2019-2688). P.G. was supported by a Senior Research Fellowship (grant 1117835) and an Investigator Grant (grant 2017070) from the National Health and Medical Research Council of Australia. T.M. and O.H. are supported by the Novo Nordisk Foundation (grant NNF18CC0034900). T.D.M. received funding from the German Research Foundation (DFG TRR296, TRR152, SFB1123, and GRK 2816/1), the German Center for Diabetes Research, and the European Research Council (ERC-CoG Trusted grant no. 101044445). L.S. is supported by the Danish Council for Independent Research, Medical Sciences (grant DFF-4004-00233), and the Novo Nordisk Foundation (grants NNF16OC0023418 and NNF18OC0032082). E.A.R. is supported by the Novo Nordisk Foundation (grant NNF18OC0034072 II), the Independent Research Fund Denmark (grant DFF-7016-00147), and the Lundbeck Foundation (grant

R233-2016-3566). M.K. is supported by the Deutsche For-
schungsgemeinschaft (DFG grant KL 3285/2-1), the German Center for
Diabetes Research (grants DZD 82DZD03D03 and 82DZD03D1Y), the
Novo Nordisk Foundation (grant NNF19OC0055192), and the Lundbeck
Foundation (grant R288-2018-78).

## Author contributions

M.K., E.A.R., B.K., L.S., T.D.M., M.H.T., A.M.-C., A.M.F., and S.H.R. con-
ceived the study or aspects of it. M.K. phenotyped the germline PanK4
KO mice. A.M.-C., with help from A.M.F., K.J., L.D., N.R.A., M.K., and
C.M.S., performed ex vivo and in vivo experiments in PanK4 mKO mice,
as well as laboratory analyses. O.H. and T.M. performed targeted
metabolite analyses. Pau. Gr. and L.S. generated AAVs to overexpress
PanK4 in mouse skeletal muscle, and L.L.V.M. and S.H.R. performed
intramuscular injections of the AAVs. C.M.S., C.S.C., and C.T.V. analyzed
human muscle samples for the effect of exercise on PanK4. A.A. and J.A.
performed global metabolite analyses. S.M.H. performed the inter-
actome analysis. R.K., T.E.J., and S.H.R. analyzed the in vivo and ex vivo
effects of exercise and contraction, and related signaling in mouse
muscle on PanK4. I.B., A.C.-S., H.L., and P.S. generated and validated
transgenic mouse models. A.S., M.J., and Pas.Go. performed analyses of
global sequencing and metabolite data. A.G. and H.P. analyzed PDH
signaling and activity in muscle. A.M.-C., A.M.F., and M.K. wrote the
manuscript. All co-authors provided input to the manuscript.

## Funding

## Competing interests

M.H.T. delivered a scientific lecture for Böhringer Ingelheim Pharma
GmbH & Co. KG (2024), AstraZeneca GmbH (2024), Lilly Deutschland
GmbH (2024) and Novo Nordisk Pharma GmbH (2024). He is co-founder
of the biotech startups Ghrelco and Bluewater Biotech (2024). As CEO
and CSO of Helmholtz Munich, M.H.T. is co-responsible for countless
collaborations of the employees with a multitude of companies and
institutions, worldwide. In this capacity, he discusses potential projects
with and has signed/signs contracts for the centers institute(s) related to
research collaborations worldwide, including but not limited to phar-
maceutical corporations like Boehringer Ingelheim, Novo Nordisk,
Roche Diagnostics, Arbormed, Eli Lilly, SCG Cell Therapy and others. As
the CEO of Helmholtz Munich, he was/is further overall responsible for
commercial technology transfer activities. M.H.T. confirms that to the
best of his knowledge none of the above funding sources or colla-
borations were involved in or had an influence on the preparation of this
manuscript. The remaining authors declare no competing interests.

## Additional information

**Supplementary information** The online version contains
supplementary material available at

Maximilian Kleinert.

**Peer review information** *Nature Communications* thanks the anon-
ymous reviewers for their contribution to the peer review of this work. A
peer review file is available.

[1]Department of Molecular Physiology of Exercise and Nutrition, German Institute of Human Nutrition (DIfE), Potsdam-Rehbruecke, Nuthetal, Germany.
[2]German Center for Diabetes Research (DZD), Munich, Germany. [3]August Krogh Section for Molecular Physiology, Department of Nutrition, Exercise and
Sports, Faculty of Science, University of Copenhagen, Copenhagen, Denmark. [4]Department of Biomedical Sciences, Faculty of Medical and Health Sciences,
University of Copenhagen, Copenhagen, Denmark. [5]Department of Forest Genetics and Plant Physiology, Swedish University of Agricultural Sciences,
Umeå, Sweden. [6]Swedish Metabolomics Centre, Umeå, Sweden. [7]Department of Anatomy and Physiology, University of Melbourne, Melbourne,
Vic, Australia. [8]Centre for Muscle Research, University of Melbourne, Melbourne, Vic, Australia. [9]Section for Cell Biology and Physiology, Department of
Biology, University of Copenhagen, Copenhagen, Denmark. [10]Metabolomics and Proteomics Core, Helmholtz Zentrum München, German Research Center
for Environmental Health, Ingolstädter Landstr. 1, Neuherberg, Germany. [11]Institute of Experimental Genetics, Helmholtz Zentrum München, German
Research Center for Environmental Health, Ingolstädter Landstraße 1, Neuherberg, Germany. [12]Department of Biochemistry, Yong Loo Lin School of Medicine,
National University of Singapore, 8 Medical Drive, Singapore, Singapore. [13]Institute of Biochemistry, Faculty of Medicine, University of Ljubljana, Vrazov trg 2,
Ljubljana, Slovenia. [14]Institute of Sports Medicine Copenhagen, Department of Orthopedic Surgery, Copenhagen University Hospital - Bispebjerg-
Frederiksberg, Copenhagen, Denmark. [15]Center for Healthy Aging, Department of Clinical Medicine, University of Copenhagen, Copenhagen, Denmark.
[16]Institute for Diabetes and Obesity, Helmholtz Diabetes Center, Helmholtz Zentrum München, Neuherberg, Germany. [17]Department of Experimental Dia-
betology, German Institute of Human Nutrition Potsdam-Rehbruecke, Nuthetal, Germany. [18]Institute of Diabetes and Regeneration Research, Helmholtz
Zentrum München, Neuherberg, Germany. [19]Division of Metabolic Diseases, Department of Medicine, Technical University of Munich, Munich, Germany.
[20]University of Potsdam, Institute of Nutritional Sciences, Nuthetal, Germany. [21]Novo Nordisk Foundation Center for Basic Metabolic Research, Faculty of
Health and Medical Sciences, University of Copenhagen, Copenhagen, Denmark. [22]Walther-Straub-Institute for Pharmacology and Toxicology, Ludwig-
Maximilian University Munich (LMU), Munich, Germany. ✉e-mail: maximilian.kleinert@dife.de

