## [Transparent Peer Review file · Nature Communications]

Pantothenate kinase 4 controls skeletal muscle substrate metabolism

Corresponding Author: Professor Maximilian Kleinert

Version 0:

Reviewer comments:

Reviewer #1

(Remarks to the Author)

The manuscript by Miranda-Cervantes et al describes pantothenate kinase 4 (PanK4) as a conserved exercise target gene that is highly expressed in skeletal muscle and dysregulated with high-fat feeding. Using a combination of germline and SkM specific PanK4 KO mice, as well as rAAV6 PanK4 overexpression constructs, the authors show that PanK4 regulates SkM fatty acid oxidation and glucose uptake in vivo and ex vivo. While this study is interesting, the experiments do not support the overall conclusions of the manuscript and additional evidence is needed. In particular, the overall results are confounded by the use of multiple skeletal muscle depots with different fiber types (slow vs fast-twitch) for critical experiments. Major and minor comments are listed below:

Major comments:

1. In Fig. 1a, the authors show that p-PanK4Ser63, is upregulated by exercise. Is the insulin-responsive phosphorylation site of PanK4 also altered with exercise or HFD feeding? What about the phosphorylation/total levels of PanK1-3?
2. The authors demonstrate that PanK4 is highly upregulated in skeletal muscle, particularly gastroc (fast twitch) and quad (fast and slow twitch). What about soleus (slow twitch), which is more oxidative, and may be more relevant for metabolic inflexibility during insulin resistance?
3. Major key experiments are conducted in different muscle depots with various fibrous types which may confound the overall conclusions of the paper, as different fiber types have various metabolic needs (i.e. slow-twitch have dense mitochondria and have high oxidative capacity, while fast-twitch muscle have a lower oxidative capacity are more glycolytic). Where possible, major experiments (ie expression and functional metabolic studies) should be conducted in the same muscle depot with the same fiber type.
4. The authors identify PanK4 as an exercise target and later go on to perform many mechanistic studies in HFD-fed mice. Can the authors comment why exercise regulates phosphorylation, while HFD feeding causes alterations in mRNA. What is the upstream signal? The manuscript would also benefit from assessing how muscle-specific PanK4 KO or mice with mutations in the phosphorylation site, respond to different exercise challenges.
5. The authors show that PDH activity was altered in the skeletal muscle of SkM-PanK4 KO mice. Was phosphorylation of PDH and/or PDK levels altered?
6. Was glucose uptake/oxidation altered in the skeletal muscle of SkM-PanK4 KO mice in vivo? Metabolic cages studies to assess RER conducted under fed/fasted conditions may provide insight.
7. Were total malonyl-CoA levels altered in SKM-PanK4 KO mice?

Minor comments:

1. Is PanK4 phosphorylation altered with HFD feeding?
2. How do the PanK4 variants alter PanK4 expression? Are they gain or loss of function/directionally related to the mouse studies? As is, these results may be better suited for the supplement.
3. Can the authors comment on the blot in Figure 1b? A description of the source of these samples is lacking. Furthermore this data shows PanK4 phosphorylation is decreased during endurance training and upregulated during sprinting or strength training. Please comment on these differences.

4. In Figure S2, tissue weight should be normalized to body weight.
5. Was fasting plasma glucose and NEFAs altered in SkM specific Pank4 KO mice?
6. Were insulin levels altered during the GTT?
7. How were acetylation/malonylation blots normalized? Ponceau Red staining should be shown.

Reviewer #2

(Remarks to the Author)

Summary

This paper examines the role of Pank4 in regulating skeletal muscle energy metabolism and metabolic flexibility. The investigators' interest in this protein stems from their initial observation that Pank4 is robustly phosphorylated in skeletal muscle in response to an acute bout of intense exercise in both rodents and humans. They then proceeded to find that the protein is expressed abundantly in skeletal muscle and heart and that SNPs of the human gene are associated with BMI and HbA1C, which together warrant further investigation of its function.

Pank4 belongs to a family of kinase enzymes that catalyze the rate-limiting step in CoA biosynthesis. Interestingly however, a recent paper by one of the leading experts in this area reported that Pank4 is a pseudoenzyme that lacks kinase activity. Thus, the function of this specific isoform remains poorly understood. Here, investigators used loss- and gain-of-function mouse models to show that Pank4 plays an important role in metabolic regulation. Muscle-specific knockout resulted in robust increases in acetyl CoA levels along with concomitant diminution of muscle fat oxidation and whole-body glucose tolerance, whereas over expression via AAV in skeletal muscle decreased acetyl CoA and increased muscle glucose uptake. The investigators proposed a working model wherein knockout of Pank4 releases a brake on synthesis of free CoA, which in turn increases acetyl CoA content resulting in feedback inhibition of PDH flux while also lowering CPT1 activity and fat oxidation by promoting malonyl CoA synthesis.

General Comments

These are very interesting and provocative findings that put Pank4 on the metabolic map and highlight a role for this protein in regulating energy metabolism and metabolic flexibility. The major weakness is the mechanistic aspect of the work; the report does not explain how Pank4 regulates acetyl CoA levels and does not provide sufficient evidence to support the working model and overall conclusions. For these reasons, the findings are viewed as intriguing but still preliminary in nature.

Major concerns

1. The study shows convincingly that Pank4 has a role in regulating acetyl CoA and muscle metabolism of glucose and lipid fuels. Still unclear is how this occurs and if/how phosphorylation alters protein function. Pank4 KO increased precursors of CoA biosynthesis (consistent with a potential role as a repressor), but free CoA levels trended downward. Presuming that KO of Pank4 increased CoA synthesis, are the investigators proposing that free CoA availability limits resting levels of acetyl CoA? Alternatively, is it possible that one of the intermediates of CoA biosynthesis serve as a signal that regulates carbon flux to acetyl CoA? In its current form, the data and proposed mechanism are confusing. Further work to clarify and strengthen the mechanistic aspect of the study would add substantial value to the report and bolster the conceptual advance.

2. In addition to the mechanism underlying the regulation of acetyl CoA, the mechanisms responsible for shifts in glucose and lipid metabolism are also unclear.

a. Fat oxidation and CPT1 inhibition. The investigators proposed that Pank4 KO resulted in malonyl CoA-mediated inhibition of CPT1, but malonyl CoA levels were not reported. Instead, lysine malonylation western blots were used as a proxy. These data were difficult to interpret because the entire blot was not shown. It appeared that the malonylated proteins affected by Pank4 KO might reside specifically in the mitochondrial matrix, which implicates a pool of malonyl CoA that would not explain CPT1 inhibition. Further assessments of malonyl CoA and beta-oxidation using ¹³C tracers, and perhaps respiration experiments performed in isolated mitochondria and/or permeabilized fibers could provide insights to substantiate the working model.

b. PDH flux. The report does not provide convincing evidence that PDH flux decreases in the KO muscles. Again, flux experiments using ¹³C tracers are needed to support the working model. Also, it would be helpful to know if PDH phosphorylation changes as well as protein abundance of the PDK enzymes, especially considering that PDK4 protein stability is known to be regulated by acetyl CoA/CoA.

c. If both PDH flux and fat oxidation are decreased, how is the acetyl CoA pool expanding? What is the source of the carbons? Considering the apparent decrease in whole-body lean mass, perhaps amino acid catabolism is increased. Assessment of pBCKDH might be informative.

d. The report shows that KO of Pank4 diminishes glucose tolerance; but the supplemental data reporting on glucose uptake and pAKT are all negative and fail to explain this phenotype.

e. The report shows that overexpression of Pank4 lowers acetyl CoA and increases glucose uptake, but does not provide

information on how this manipulation affected fat oxidation, glucose oxidation, PDH activity, and malonyl CoA.

Minor comments

- The free CoA levels are an important outcome measure and should be shown in the main figures.
- What was the rationale for subjecting mice to glucose injections prior to collecting muscles for metabolomics?
- Given that the story starts with an exercise experiment, it was surprising that the experiments to characterize Pank4 KO mice did not include an exercise tolerance test. Regardless of the outcome, this would be a nice experiment to include.
- The study should include measures of whole-body energy expenditure and RER during feeding/fasting, which is typically used as a key readout of metabolic flexibility.
- It would be helpful if the acylcarnitine and acyl CoA data were also provided in table format.
- The feeding status and length of fast prior to each experiment should be clearly noted in the figure legend.

Version 1:

Reviewer comments:

Reviewer #1

(Remarks to the Author)

The authors have provided additional evidence to support the role of PanK4 in modulating muscle substrate utilization. While the additional experiments and data have substantially strengthened some of their conclusions, they also lead to additional questions. Even with the revisions, unfortunately, the mechanistic work remains preliminary and difficult to follow.

Additional comments below:

1. The authors have now nicely demonstrated the abundance of PanK4 across different skeletal muscle beds by Western blotting in new Fig 1F. It would be helpful if the quantification of Pank1 and 2 were also added to the main figure, so readers can easily see how the different isoforms are expression across muscle beds.
2. It would be helpful if the authors could include whether the analyses were done in human or mouse tissue (male or female), which muscle depot was used, and whether the experiments were conducted in fed/fasted states to aid in readability. Not all of this information is included within all figure legends and main text, making it difficult to compare findings across different experimental settings.
3. As mentioned in my initial (minor) comments, the SNP data is rather preliminary. How does the directionality of the variant correlate with the findings in mice? This might be better suited for the supplement.
4. I commend the authors for performing metabolic cage studies and realize that it is difficult to pick up changes in whole-body substrate utilization using this technique. Fasting/refeeding experiments can often give a bigger window to see differences in whole body substrate utilization (especially during the initial transition to the fasting state- see PMID: 22517326, for example). From the RER plot, it looks like during the initial part of the fast the KO mice utilize more carbs (RER closer to 1)—this assumes the inset though has the correct alignment of when the fast actually started. If the inset is correct- was carbohydrate utilization significantly increased during this time? Average activity is also increasing in the KO mice, which seems relevant to discuss in the results given the effect of PanK4 KO on exercise.
5. Were levels of fasting/fed plasma glucose, NEFAs or BOHB altered in the muscle specific KO mice?
6. As presented, it is often unclear what groups are statistically significant in the figures. It would be helpful to always present the data in the same way (KO next to controls within fed or fasted state).
7. The authors now nicely show that PanK4 regulates glucose uptake during exercise in vivo but there are no experiments showing that PanK4 regulates glucose utilization in vivo (only PDH activity is measured ex vivo in KO mice and glucose uptake is only assessed in the O/E model; further pPDH is not changing in vivo). It is further still unclear how PanK4 regulates acetyl-CoA levels. Further, if glucose and fat utilization is down, what substrate is the muscle using?

Reviewer #2

(Remarks to the Author)

The investigators have performed a substantial amount of additional work and have provided several new data sets. While the manuscript has been improved, some of the most important gaps that had been identified previously were not sufficiently satisfied.

Major comments

1. Considering that the report concludes that PanK4 controls muscle oxidation of both fatty acid and glucose, reviewers asked how the loss of PanK4 in muscle affects energy expenditure, substrate utilization, and exercise performance at the whole-body level, as well as muscle PDH phosphorylation and PDK4 stability and abundance. To their credit, investigators produced and shared several new analyses to address these questions. The results of all those experiments and analyses

were surprisingly negative. While negative data can be critically important for developing a clear picture of function and mechanism; in aggregate, the negative outcomes here raise strong doubt that loss of PANK4 results in diminished PDH flux, reduced glucose oxidation, and inefficient energy metabolism - as implied in the title, abstract, and narrative. There is simply no evidence to support these aspects of the working model.

2. Figure 4d shows that insulin-stimulated 2DOG uptake is decreased in EDL, whereas basal levels were unchanged, suggesting that the KO EDL muscles exhibit diminished insulin action at the level of glucose uptake, without affecting insulin signaling. By contrast, neither basal nor insulin-stimulated 2DOG uptake were affected in Pank4KO soleus muscles. Also noteworthy, Figure 2 shows that KO soleus muscles had lower oxidation rates of ¹⁴C fatty acid when assayed *ex vivo*, but only when the muscles were stimulated to contract. Therefore, the impact of Pank4 on both glucose uptake and fat oxidation appears to depend on fiber type and context, even though acetyl CoA was elevated in all states tested. Similar assays using radio-labeled glucose could be performed in both soleus and EDL to determine if changes in glucose oxidation rates are indeed evident under these same conditions.

In sum, the finding that loss of Pank4 results in diminished uptake of 2DOG into muscles under some (but not all) circumstances does not establish perturbed PDH flux and/or glucose oxidation. While the investigators' assumption might be correct, this claim remains entirely speculative based on the data provided. Also at odds with this suggestion is that acetyl CoA, malonyl CoA and citrate levels are elevated in mKO muscles, which typically occurs when PDH flux increases.

3. The investigators made a strong and commendable effort to assay malonyl CoA levels. When taken together with the other findings, these new data do provide solid evidence that KO of Pank4 results in diminished fat oxidation, likely due in part to malonyl CoA-mediated inhibition of CPT1a.

4. Considering that KO of Pank4 did not affect free CoA levels and markedly increased acetyl CoA, it remains unclear whether or not this protein actually regulates CoA synthesis and why acetyl CoA increases. The results implying that Pank4 binds to BDH1 are quite intriguing; however, these findings are impossible to interpret without further assessment of muscle ketone metabolism.

5. The report shows that overexpression of Pank4 lowers acetyl CoA and increases glucose uptake, but does not provide information on how this manipulation affected fat oxidation and glucose oxidation.

Minor comments

The title states that Pank4 controls efficient substrate use. What is meant by "efficient"? What is the evidence of inefficiency?

If muscle glucose uptake is diminished in KO mice during exercise, but exercise tolerance and performance are unaffected; how are the mice supporting muscle contraction? Is there any evidence that use of ketones and/or muscle glycogen are increased in KO mice? Was muscle lactate production different?

Considering that Pank4 localizes to the IMS and appears to elevate the malonyl CoA pool that regulates CPT1 activity, have the authors considered the possibility that Pank4 is regulating the acetyl CoA pool that resides outside the mitochondria, rather than the matrix pool?

In the instances when differences between groups do not reach statistical significance, results should be discussed as such (e.g. PDH activity, basal fat oxidation).

Summary

The revised manuscript now provides solid evidence that loss of Pank4 results in decreased fat oxidation, most likely due to increased malonyl CoA levels and diminished CPT1 activity. Still lacking is any evidence that PDH flux and glucose oxidation are "gridlocked". Still unknown is why acetyl CoA increases; why insulin-stimulated glucose uptake is diminished in some but not all muscles; if CoA biosynthesis is affected; and if phosphorylation of Pank4 alters protein activity. At the whole-body level, energy balance, substrate use, exercise performance, and responses to high fat feeding were all surprisingly unaffected, suggesting that the overall physiological impact of this protein on energy metabolism is limited to glucose response during a GTT. While the work is novel and the emergent role of Pank4 in regulating acetyl CoA and mal CoA is very intriguing, the findings remain largely descriptive and some of the conclusions are not convincingly supported by the data provided. Nonetheless, the novelty and potential value of the science are recognized. The authors are therefore strongly encouraged to temper speculation with regard to mechanisms and therapeutic implications (which seem premature), and further acknowledge the limitations of the work at this stage.

Version 2:

Reviewer comments:

Reviewer #1

(Remarks to the Author)

I commend the authors for performing additional work to answer key questions raised by both reviewers and toning down their conclusions during this revision process. While I acknowledge the value in identifying Pank4 as an exercise target and

believe the work is important and novel, unfortunately the major concerns that were raised by both reviewers are still not sufficiently addressed. The lack of consistency in terms of the whole-body substrate utilization and mechanistic work (circulating substrates, glucose uptake, metabolomics, PDH activity) with different stimuli (exercise, contraction, HFD) and muscle fiber types (oxidative vs glycolytic) makes it hard to draw conclusions about the mechanism(s) by which Pank4 alters substrate utilization, and unfortunately, lessens my enthusiasm for this work.

Reviewer #2

(Remarks to the Author)

The revised title and narrative are now much better aligned with the data presented. The work is significant and provides important new insights into the role of Pank4 in regulating muscle metabolism. Overall, the findings are novel and provocative, and likely to attract strong interest from the metabolic research community.

Dear Reviewers,

We would like to express our gratitude to the reviewers for their thorough appraisal of our work and their excellent questions and suggestions. Your inputs have significantly improved the manuscript.

In our revised submission, we have included several additional studies and analyses. We have added human data on the regulation of p-PanK4 Ser63 by different exercise modalities. We provide new information on the expression of PanK4 and its family members in different organs and in different skeletal muscles. We investigated the recently identified T406 phosphorylation site on PanK4, examining its regulation by exercise (and insulin) in human muscle. We explored how the loss of PanK4 in muscle affects whole-body energy expenditure and substrate utilization, muscle PDH phosphorylation, as well as exercise performance. We examined muscle glucose use in response to treadmill running and found that **PanK4 regulates glucose uptake into skeletal muscle in response to exercise**. We have performed whole-genome RNA sequencing analysis establishing a resource of transcriptional regulatory network downstream of PanK4 signaling. We also investigated the signaling mechanisms that regulate p-PanK4 Ser63 during exercise in various additional transgenic mouse models and pharmacological approaches. Our findings show that contractions induce Ser63 phosphorylation on PanK4 likely via a calcium-dependent mechanism, while AMPK and stretch-related signaling are ruled out. We also explored PanK4 binding partners using two immuno-purification strategies, mass spectrometry, and PanK4 knockout samples as controls. This has provided a high-quality hypothesis-generating resource for further research into PanK4 biology.

We also **succeeded in determining muscle malonyl-CoA levels**, a key measurement requested by both reviewers and one we were keen to determine ourselves. This task proved challenging due to the low abundance of malonyl-CoA in skeletal muscle, signal suppression by co-eluting components in mass spectrometry, and its poor stability in solution. Initially, these challenges resulted in insufficient signal intensity, with peak areas near the noise level and below the calibrated range, making accurate measurement difficult. For these reasons, we were unable to provide these data in our first submission. However, by meticulously optimizing extraction procedures, workflows, choice of column, and using a new mass spectrometer with higher sensitivity, we succeeded in measuring malonyl-CoA, and, excitingly, **found levels of malonyl-CoA to be elevated in muscle lacking PanK4**. These results support the hypothesis that elevated acetyl-CoA leads to increased malonyl-CoA, explaining reduced fatty acid mitochondrial import and oxidation in the absence of PanK4. We are excited to be able to show these critical data in our revised version and we hope the reviewers appreciate the considerable effort invested in overcoming these technical challenges and the future research perspectives for others in using our optimized mass spectrometry-based methodology development.

We trust that the reviewers recognize our extensive efforts over the past months and share our enthusiasm for the discovery of **the pivotal role of PanK4 in skeletal muscle metabolism**.

Please find our point-by-point replies below.

Reviewer # 1 Major comments

1. In Fig. 1a, the authors show that p-PanK4Ser63, is upregulated by exercise. Is the insulin-responsive phosphorylation site of PanK4 also altered with exercise or HFD feeding? What about the phosphorylation/total levels of PanK1-3?

Thank you for these stimulating questions.

Regarding the insulin-responsive site of PanK4, as of now, no antibodies specifically targeting on PanK4 (Thr406), the insulin-responsive site reported by Dibble et al. (PMID: 35896750), are available. The authors in that paper immunoprecipitated PanK4 and used a pan-phospho Akt substrate antibody to evaluate insulin-responsive phosphorylation of PanK4. Notably, a recent study by Needham et al. (PMID: 34857927), which looked at exercise and insulin signaling in human skeletal muscle by phosphoproteomic analysis, indicates that PanK4 T406 phosphorylation increases with insulin in human skeletal muscle but not in response to exercise. Conversely, the same study found that PanK4 Ser63 phosphorylation is responsive to exercise

(aligning with our observations) but not to insulin. Regarding the inquiry about the "phosphorylation/total levels of PanK1-3", our investigation is again constrained by the absence of specific antibodies. Nonetheless, three phosphoproteomic studies focused on exercise signaling in muscle (PMID: 34857927, PMID: 26437602, PMID: 35882232) have reported phosphorylation sites only on PanK2 and PanK4 in human skeletal muscle, whereas sites on PanK1 and PanK3 were not detected. This likely reflects a higher abundance of PanK2 and PanK4 in muscle tissues, which we further investigated and we able to confirm with new data from mouse muscle (see new Figure 1E,F). In the phosphoproteomic investigations, phosphorylation of PanK4 S63 consistently increased with exercise regardless of modality and duration. In fact, our new Western blotting analyses (new Figure 1C) further corroborate these findings, demonstrating a significant induction of phosphorylated PanK4 S63 in muscle following endurance, sprint, and resistance exercise, reinforcing the conclusion that PanK4 is a robust and universal exercise-responsive target in muscle across various exercise modalities. PanK2 phosphorylation, in contrast, did not respond to exercise, with the exception being a marginal increase (+14%) in PanK2 S45 following sprint exercise in humans (PMID: 34857927), with no effects on PanK2 S45 following endurance- and resistance-type exercises.

Summary of three phosphoproteomic studies focused on exercise signaling in muscle (PMID: 34857927, PMID: 26437602, PMID: 35882232)

Furthermore, we provide additional data on PanK4 abundance, relative to other organs and also relative to the other PanKs. Based on new RNAseq data, we now show that in muscle total mRNA abundance of *Pank4* is 3-, 7-, and 9-fold higher than that of *Pank2*, *Pank3*, and *Pank1*, respectively (new Fig 1E), supporting the notion that PanK4 is the most abundant PanK in muscle. New Western blotting results support this, showing that PanK4 expression is significantly higher in skeletal muscle than in organs such as the liver; and that the expression levels of PanK4 across different skeletal muscle types are relatively consistent. In contrast, PanK1 and PanK2 exhibit far greater abundance in most non-muscle tissues, compared to skeletal muscle.

In summary, to address the reviewer's question, we provide new data on PanK4 phosphorylation in response to exercise, and we have added data showing that PanK4 is the most abundant PanK in skeletal muscle.

2. The authors demonstrate that PanK4 is highly upregulated in skeletal muscle, particularly gastroc (fast twitch) and quad (fast and slow twitch). What about soleus (slow twitch), which is more oxidative, and may be more relevant for metabolic inflexibility during insulin resistance?

Thank you for this comment. Following the reviewer's suggestion, we now examined the abundance of PanK4 in different skeletal muscles and other tissues through Western blotting (new Figure 1F and Figure S1E,F). It can be concluded that PanK4 expression is higher in skeletal muscle compared to other organs. We observed no marked differences among different muscles including soleus. Overall, it appears that PanK4 is ubiquitously expressed but is enriched in striated muscle.

3. Major key experiments are conducted in different muscle depots with various fibrous types which may confound the overall conclusions of the paper, as different fiber types have various metabolic needs (i.e. slow-twitch have dense mitochondria and have high oxidative capacity, while fast-twitch muscle have a lower oxidative capacity are more glycolytic). Where possible, major experiments (ie expression and functional metabolic studies) should be conducted in the same muscle depot with the same fiber type.

This is a fair point, and some inconsistencies have arisen from the collaborative nature of this project. However, key functional experiments, such as insulin- and exercise-stimulated glucose uptake, have been assessed in multiple skeletal muscle types also representing the most polarized fiber type differences within mouse

muscles such as EDL (the most glycolytic muscle in a mouse) and soleus (the most oxidative muscle in a mouse). Additionally, key outcomes like increased acetyl-CoA levels were detected by non-targeted metabolomics in TA (very glycolytic) and whole gastrocnemius (more mixed between glycolytic and oxidative fibers) muscles. These findings were further validated by targeted analyses in gastrocnemius muscle in the fasted, fed, and exercised states. While we agree that the question of fiber-type dependent effects is interesting, we believe that demonstrating effects in different muscle types actually strengthens our findings. Also, our data on the expression of PanK4 in various muscle depots on the entire continuum of fiber type expressions as well as measurement of metabolic disturbances in PanK4 knock-out muscles from different depots do not imply strong fiber type specific differences. However, we agree with the reviewer that unraveling perhaps more nuanced fiber-type dependent mechanisms of how PanK4 is regulated and what PanK4 regulates will be a valuable undertaking in the future, which we briefly mention now in the discussion to highlight this to the reader (p 10, ln 30-33).

4. The authors identify PanK4 as an exercise target and later go on to perform many mechanistic studies in HFD-fed mice. Can the authors comment why exercise regulates phosphorylation, while HFD feeding causes alterations in mRNA. What is the upstream signal? The manuscript would also benefit from assessing how muscle-specific PanK4 KO or mice with mutations in the phosphorylation site, respond to different exercise challenges.

While we showed that high-fat diet (HFD) feeding leads to a reduction in *Pank4* mRNA, it is important to note that all subsequent data in our study were obtained from mice on a chow-fed diet. Therefore, our focus was not specifically on HFD-related effects.

The question about exercise challenges is excellent. We therefore undertook extensive efforts to address this within a reasonable timeframe. We have several updates. First, muscle phospho-PanK4 S63 increases with treadmill running, *in situ* contractions, and *ex vivo* contractions in mice (Fig S7 and S8), indicating that contractions alone are sufficient to increase this phosphorylation site. This rules out the possibility that the increase is solely driven by a hormonal response associated with exercise. Using various additional transgenic mouse models and pharmacological approaches, we have also ruled out AMPK and stretch signaling as the upstream signaling pathways mediating p-PanK4 during contractions, while identifying that calcium-dependent mechanisms likely play a role (Fig 8).

Using our PanK4 mKO mice, we found that while maximal and endurance running capacities were normal, muscle glucose uptake in response to treadmill running is impaired (Fig 4), indicating that **PanK4 is crucial for normal muscle glucose uptake during exercise**. Impaired exercise-induced glucose uptake was associated with increased muscle acetyl-CoA and citrate levels in PanK4 mKO mice (Fig 4H, Fig S7B). These new data contribute to the overall evidence that PanK4 is essential for normal muscle acetyl-CoA homeostasis and efficient metabolic substrate use during physiological challenges to metabolic flexibility.

While the specific mechanisms are still under investigation, the significant role of PanK4 in muscle metabolism, given its high abundance in skeletal muscle, is extremely exciting. This discovery not only raises additional questions but also suggests potential therapeutic avenues for metabolic diseases.

5. The authors show that PDH activity was altered in the skeletal muscle of SkM-PanK4 KO mice. Was phosphorylation of PDH and/or PDK levels altered?

This is an excellent point. Following this suggestion, we assessed phosphorylation of PDH and abundance of PDKs by Western blotting (Fig S5E-L). There was no difference in p-PDH and PDK levels between PanK4 WT and PanK4 mKO mice, suggesting that perhaps other post-translational modifications (e.g., acetylation or succinylation) regulate PDH activity.

6. Was glucose uptake/oxidation altered in the skeletal muscle of SkM-PanK4 KO mice *in vivo*? Metabolic cages studies to assess RER conducted under fed/fasted conditions may provide insight.

Following the reviewer's suggestion, we performed metabolic cage studies in PanK4 WT and PanK4 mKO mice exposed to chow, fasting, and refeeding, as well as short-term HFD feeding (Fig S3). These data revealed no significant difference in energy expenditure and substrate utilization. While we agree with the reviewer that

this approach could have provided important physiological insights, it is important to note that mouse metabolism is more liver- and brown adipose tissue-centric (especially at 23°C) than that of humans, which could mask effects. In agreement, there are examples of key muscle metabolic enzymes not having an effect on whole-body energy expenditure and energy substrate utilization. For instances, mice lacking both AMPK α 1 and - α 2 in skeletal muscle exhibited similar whole-body metabolism on both chow and HFD, while ex vivo muscle fatty acid oxidation was impaired (PMID: 25609422). Similarly, while germline deletion of Akt1 resulted in increased whole-body oxygen consumption (and protection from diet-induced obesity), muscle-specific Akt1 deletion had no effect (PMID: 22037765).

7. Were total malonyl-CoA levels altered in SKM-Pank4 KO mice?

This is an excellent question, and we were eager to address ourselves with the expertise of world-renowned metabolomics experts. Analyzing malonyl-CoA proved extremely challenging, because:

- 1) Malonyl-CoA is a low-abundance metabolite in skeletal muscle
- 2) Its sensitivity is lower compared to other metabolites analyzed in the same run
- 3) Co-eluting matrix components with malonyl-CoA during hydrophilic interaction chromatography suppress its signal in mass spectrometry-based analysis.
- 4) Malonyl-CoA has poor stability in solution, degrading relatively quickly once dissolved, as shown in Fig. A below.

These challenges contributed to our initial inability measure malonyl-CoA due to insufficient signal intensity, with peak areas close to the noise region and below the calibrated range. However, by optimizing the extraction protocol, minimizing the time from extraction to measurement, altering the column length, and using newer, more sensitive instrumentation, we successfully determined malonyl-CoA levels in muscle. Notably, internal isotopically labeled standard (malonyl-CoA- $^{13}\text{C}_3$) was used to quantify malonyl-CoA accurately.

Using our optimized detection approach, we first observed a trend for higher malonyl-CoA levels in muscle lacking Pank4 in fasted versus fed mice (Fig 3F). To conclusively determine if malonyl-CoA is increased, we analyzed malonyl-CoA in two muscles from a larger new cohort of fed Pank4 WT and Pank4 mKO mice (n = 12). Samples were collected in our lab in Germany and sent to Sweden for analysis. The personnel dissecting the muscles and analyzing them were blinded to the genotype. These efforts paid off, as the new data showed a highly significant ($p < 0.0001$) **two-fold increase in malonyl-CoA levels in muscles lacking Pank4** compared to control muscles (Fig 3G).

This is a critical finding, as malonyl-CoA is a key allosteric inhibitor of CPT1, an integral part of the carnitine shuttle that facilitates the mitochondrial import of long-chain fatty acids. These data, together with the observed higher total lysine malonylation levels, indicate that higher acetyl-CoA results in higher malonyl-CoA levels, which likely explain the observed decrease in fatty acid oxidation.

Our detailed workflow for detecting malonyl-CoA is carefully described in the methods section to serve as a resource for others measuring this key metabolite in muscle. We hope the reviewer can appreciate the effort that went into generating these important data.

Reviewer # 2 Major comments

1. The study shows convincingly that Pank4 has a role in regulating acetyl CoA and muscle metabolism of glucose and lipid fuels. Still unclear is how this occurs and if/how phosphorylation alters protein function. Pank4 KO increased precursors of CoA biosynthesis (consistent with a potential role as a repressor), but free CoA levels trended downward. Presuming that KO of Pank4 increased CoA synthesis, are the investigators proposing that free CoA availability limits resting levels of acetyl CoA? Alternatively, is it possible that one of the intermediates of CoA biosynthesis serve as a signal that regulates carbon flux to acetyl CoA? In its current form, the data and proposed mechanism are confusing. Further work to clarify and strengthen the mechanistic aspect of the study would add substantial value to the report and bolster the conceptual advance.

We are thrilled to hear that the reviewer agrees we have convincingly demonstrated PanK4's significant role in regulating acetyl-CoA and the metabolism of glucose and lipid fuels in skeletal muscle. The reviewer's comments make it clear that we have opened some promising avenues related to the underlying mechanisms, which we are eager to explore further with the scientific community. Additionally, from a functional standpoint, we believe that the scientific community will share the reviewer's excitement about PanK4 being a major new player in muscle metabolism. The reviewer has raised overall many excellent but also challenging points. While we did not succeed in addressing all requests, we hope that our extensive overall efforts will be appreciated and deemed sufficient, leading to a clearer picture on the PanK4 function.

2. In addition to the mechanism underlying the regulation of acetyl CoA, the mechanisms responsible for shifts in glucose and lipid metabolism are also unclear.

a. Fat oxidation and CPT1 inhibition. The investigators proposed that PanK4 KO resulted in malonyl CoA-mediated inhibition of CPT1, but malonyl CoA levels were not reported. Instead, lysine malonylation western blots were used as a proxy. These data were difficult to interpret because the entire blot was not shown. It appeared that the malonylated proteins affected by PanK4 KO might reside specifically in the mitochondrial matrix, which implicates a pool of malonyl CoA that would not explain CPT1 inhibition. Further assessments of malonyl CoA and beta-oxidation using ^{13}C tracers, and perhaps respiration experiments performed in isolated mitochondria and/or permeabilized fibers could provide insights to substantiate the working model.

These are all fair points which were all considered extensively.

First, we apologize for over-cropping the blot. We have now tried to remedy this and also now indicate molecular weight markers. We would argue that it is difficult to deduce which proteins are malonylated based on their molecular size. It is also worth noting that while the higher molecular-weight bands are overall fainter (higher molecular weight proteins transfer slower from the gel to the membrane, making it difficult to deduce total protein abundance based on the band intensity) the genotype difference persists there as well.

Although we do recognize that ^{13}C flux analyses have provided some valuable insights into muscle metabolic fluxes in mice in a few papers, it is still a very limited numbers of laboratories worldwide that has this methodology established and it is not really possible to rederive our mice at these institutions and perform such analysis within a reasonable timeframe of revising the manuscript. Still, we believe that in regards to beta-oxidation our ^{14}C -palmitate data (Fig 2F) convincingly demonstrate that PanK4 is required for efficient exogenous LCFA-oxidation. We discussed extensively the reviewer's idea to direct resources towards the suggested respiration experiments, but decided against it because both the isolated mitochondria and/or permeabilized fibers the endogenous metabolite milieu, that we think is key for the defects - would be absent.

Regarding **the detection of muscle malonyl-CoA levels**, this is an excellent point and we were eager to address this with the expertise of world-renowned metabolomics experts. Analyzing malonyl-CoA proved extremely challenging, because:

- 1) Malonyl-CoA is a low-abundance metabolite in skeletal muscle
- 2) Its sensitivity is lower compared to other metabolites analyzed in the same run
- 3) Co-eluting matrix components with malonyl-CoA during hydrophilic interaction chromatography suppress its signal in mass spectrometry-based analysis.
- 4) Malonyl-CoA has poor stability in solution, degrading relatively quickly once dissolved, as shown in Fig. A below.

These challenges contributed to our initial inability measure malonyl-CoA due to insufficient signal intensity, with peak areas close to the noise region and below the calibrated range. However, by optimizing the extraction protocol, minimizing the time from extraction to measurement, altering the column length, and using newer, more sensitive instrumentation, we successfully determined malonyl-CoA levels in muscle. Notably, internal isotopically labeled standard (malonyl-CoA- $^{13}\text{C}_3$) was used to quantify malonyl-CoA accurately.

Using our optimized detection approach, we first observed a trend for higher malonyl-CoA levels in muscle lacking PanK4 in fasted versus fed mice (Fig 3F). To conclusively determine if malonyl-CoA is increased, we analyzed malonyl-CoA in two muscles from a larger new cohort of fed PanK4 WT and PanK4 mKO mice (n = 12). Samples were collected in our lab in Germany and sent to Sweden for analysis. The personnel dissecting the muscles and analyzing them were blinded to the genotype. These efforts paid off, as the new data showed a highly significant ($p < 0.0001$) **two-fold increase in malonyl-CoA levels in muscles lacking PanK4** compared to control muscles (Fig 3G).

This is a critical finding, as malonyl-CoA is a key allosteric inhibitor of CPT1, an integral part of the carnitine shuttle that facilitates the mitochondrial import of long-chain fatty acids. These data, together with the observed higher total lysine malonylation levels, indicate that higher acetyl-CoA results in higher malonyl-CoA levels, which likely explain the observed decrease in fatty acid oxidation.

Our detailed workflow for detecting malonyl-CoA is carefully described in the methods section to serve as a resource for others measuring this key metabolite in muscle. We hope the reviewer can appreciate the effort that went into generating these important data.

b. PDH flux. The report does not provide convincing evidence that PDH flux decreases in the KO muscles. Again, flux experiments using ^{13}C tracers are needed to support the working model. Also, it would be helpful to know if PDH phosphorylation changes as well as protein abundance of the PDK enzymes, especially considering that PDK4 protein stability is known to be regulated by acetyl CoA/CoA.

The reviewer makes a great point about PDK4 stability being regulated by acetyl CoA/CoA. We therefore assessed phosphorylation of PDH and PDK abundance by Western blotting (these data can be found in Fig S5E-L). There was no difference in p-PDH and PDK levels between PanK4 WT and PanK4 mKO mice, suggesting that other post-translational modifications (e.g., acetylation or succinylation) likely regulated PDH activity.

c. If both PDH flux and fat oxidation are decreased, how is the acetyl CoA pool expanding? What is the source of the carbons? Considering the apparent decrease in whole-body lean mass, perhaps amino acid catabolism is increased. Assessment of pBCKDH might be informative.

The reviewer makes an astute observation, one that has been on our minds: where is the excess acetyl-CoA coming from? As per the reviewer's suggestion, we assessed both total BCKDH and pBCKDH but observed no dysregulation in PanK4 mKO muscle (Fig S9). We also performed RNA-sequencing analysis of muscles from PanK4 WT and mKO, and specifically examined mRNA of genes encoding enzymes involved in the regulation of BCAA catabolism, however, we failed to identify major alterations (Fig S9). Re-examining our metabolomics data, we do not see a coordinated increase in BCAA-related metabolites that could drive the increase in acetyl-CoA. Thus, we currently conclude that BCAA catabolism likely does not contribute.

The question remains regarding the source of increased acetyl-CoA; whether it results from an oversupply from certain metabolic processes or an underutilization by acetyl-CoA-consuming pathways. Notably, we found strong evidence for an interaction between PanK4 and BDH1 (new Fig S10, discussed more below), a key enzyme in the pathway facilitating the conversion of the exogenous ketone body beta-hydroxybutyrate into acetyl-CoA within the cell. Whether PanK4 regulates BDH1 is speculative at this point, but we make the reader aware of this intriguing finding (p 9, ln 1-3).

Another interesting observation was that mRNAs of several enzymes mapping to the polyamine/spermidine pathway (e.g., Odc1, Sms, and Smox) were reduced in PanK4 mKO muscle. It has been suggested that activation of the polyamine/spermidine pathways consumes acetyl-CoA to generate metabolites such as acetyl-spermine, diacetyl-spermine, and acetyl-spermidine. Therefore, an underutilization of this pathway, as indicated by the RNAseq data, could result in reduced consumption of acetyl-CoA and hence contribute to the higher acetyl-CoA levels. This, of course, is also highly speculative for now, but we wanted to make the reviewer aware of this.

d. The report shows that KO of Pank4 diminishes glucose tolerance; but the supplemental data reporting on glucose uptake and pAKT are all negative and fail to explain this phenotype.

With the greatest respect, but the glucose uptake data are not negative. We do demonstrate that insulin-stimulated **glucose uptake into the predominant glycolytic EDL muscle is impaired (Fig. 4d)**, indicating that there is a defect, particularly in glycolytic muscle fibers (which is the predominant fiber type in mice), likely contributing to glucose intolerance on the whole-body level. The regulation of insulin-stimulated glucose uptake in muscle is complex. Briefly, it depends on 1) glucose/insulin delivery to the muscle, 2) transport across (via transporters) the cell membrane (and in principle also endothelial membranes), and 3) intracellular metabolism in order to keep a gradient of glucose for uptake across sarcolemma. The proposed mode-of-action in the present study is a mechanism via intracellular metabolism which can occur independently of Akt. For example, we report a buildup in citrate levels and citrate has been proposed to be an allosteric inhibitor of glycolysis enzymes (e.g., PMID: 1831450) and we also report impaired PDH activity that similarly will impair glycolytic flux. With regards to the importance of glycolytic flux, it was recently shown that increasing glycolytic flux in muscle, notably by pharmacological activation of PDH, increases glucose uptake independently from insulin signaling (e.g. no effect on Akt S473 and T308 phosphorylation (PMID: 38608261)). In this line, regulation of glycolysis has also recently been established as one of the main culprits of insulin resistance in muscle (PMID: 35021042; 38329473; 26342081) and upregulating glycolysis is shown to reverse insulin resistance (PMID: 37494090; 38329473). This demonstrates that insulin-regulated glucose utilization is complex but can definitely be driven or altered by signaling and/or metabolic alterations independent of Akt. To support the point that Akt is not the sole regulator of muscle glucose uptake, it was observed by others that the knockout of both Akt1 and Akt2 in skeletal muscle resulted in largely normal insulin-stimulated glucose uptake and exhibited normal glucose tolerance (PMID: 31444134). Accordingly, an accumulating body of evidence also emphasize lipid-induced impairment in insulin-stimulated glucose uptake into skeletal muscle independently of regulation of the proximal insulin signaling cascade (PMID: 22851577; 20956497; 17652214; 15001626).

e. The report shows that overexpression of Pank4 lowers acetyl CoA and increases glucose uptake, but does not provide information on how this manipulation affected fat oxidation, glucose oxidation, PDH activity, and malonyl CoA.

That is a fair point, and we would have liked to provide all the additional information mentioned. Measurements of fat, glucose oxidation, PDH activity, and metabolites would have required several additional mouse cohorts for which we had insufficient amounts of vector. However, what we did accomplish is using the lysates from these experiments to explore PanK4 binding partners. The flag-tag on PanK4 facilitated the isolation of tagged PanK4, but we also pulled down endogenous PanK4, creating two independent data sets. Similar pulldowns were performed in muscle tissues from PanK4 knockout mice to control for nonspecific binding. We examined interaction partners using mass spectrometry. We are now reporting this interactome as a resource for future efforts to better understand PanK4 function (Fig S10). One observation stood out to us: BDH1 interacts with both flag-tagged and endogenous PanK4. BDH1 converts beta-hydroxybutyrate (entering muscle via the MCT1 transporter) into acetoacetate. This is subsequently converted into acetoacetyl-CoA and ultimately acetyl-CoA, suggesting an increase in acetyl-CoA due to higher utilization of exogenous ketone bodies. Interestingly, it has recently been shown by the Muoio group that that SkM BDH1 also optimizes fat oxidation (PMID: 38325337). This exciting link demonstrates why we believe that these data will be a valuable resource to generate new hypotheses and further dissect PanK4's function in the future.

Dear Reviewers,

We appreciate your fast turnaround and continued input to our work. Overall your comments are constructive and help to improve the quality of the manuscript. In addition to a specific point-by-point response to each of your comments below, first there are two overall emerging meta-points of criticism arising that we wanted to address.

Point 1: the whole-body phenotype is disappointing and highlights a limited role for PanK4 in muscle metabolism.

With all respect, this is not a scientifically nuanced interpretation. Mice are more liver-centric organisms compared to humans. While skeletal muscle is the primary site of glucose disposal in humans, it is the liver in rodents. This is evident from the fact that the relative liver weight is approximately 2.5 times greater in rats and mice than in humans, and glycogen concentrations per unit of tissue mass in human muscle are 20-fold higher than in mouse muscle. This might explain why famous muscle metabolic signaling enzymes, such as AMPK, Akt, Rac1, TBC1D1/4, which have been investigated for years, have very well-established crucial roles in metabolic regulation in muscle and on whole-body level in health and metabolic disease, and cited thousands of times in the context of muscle metabolism, have often no or very limited effects on whole-body substrate utilization, when deleted specifically in muscle. Moreover, in terms of whole-body physiological impact, the marked glucose intolerance observed by us should not be understated. It is also feasible that the impact of PanK4 on metabolism is fiber type dependent with effects on glucose uptake and FAOX in glycolytic and oxidative fibers, respectively. In such a scenario, the effects on whole-body substrate utilization could cancel out.

Point 2: the mechanistic depth is inadequate

Yes, there are many unanswered questions, but is this not the mark of a new research field? Mechanistic work is very difficult to perform in mature organs in in vivo experiments. One can envision creating multiple knockout mouse models but these take time and also have serious caveats regarding their interpretation. It should also be noted that the very interesting 2022 Nature paper¹ also left several questions unanswered. For example, a new insulin responsive and Akt-dependent phosphorylation site on PanK4 (T406) was discovered, however, the function of this phosphorylation site was not clarified. Yes, phospho-mutants (aiming to turn this site “” or “off”) seemed to moderately modulate CoA levels in cancer cells (with constitutive active Akt and KO of PanK4) but this occurred together with marked differences in cell proliferation, which makes it difficult to interpret these results. Importantly to note, the authors showed that these phospho-mutants (T406A or T406E) had no impact on the phosphatase activity of PanK4 (Figure S10b in 2022 Nature¹):

[REDACTED]

So, in the end, the mechanism for what this phospho-site does remains to be discovered. Also, while the authors very elegantly demonstrated that PanK4 is negative regulator of CoA synthesis (in cancer cells), the metabolic consequences (i.e., effects on glucose or lipid utilization) were not assed. The authors also observed that PanK4 regulates acetyl-CoA, but the mechanism for this increase or its subcellular localization were also not investigated. This is by no means meant to diminish the impact of this very important publication, but it highlights that we are at the very beginning of a new research topic which needs the research community as a whole to tackle all these emerging questions. We strongly believe that our data showing that PanK4 is an exercise target, showing that PanK4 is abundant in muscle, showing that whole-body KO regulates body size and IGF-1, showing that muscle PanK4 regulates acetyl-CoA and malonyl-CoA, showing that PanK4 regulates glucose and lipid metabolism, and showing that increasing PanK4 in sufficient to increase glucose uptake is highly relevant, needs to be shared now with the scientific community and undoubtedly clears the bar for publication in *Nature Communication*.

Lastly, we do, however, agree with the reviewers that the link between PDH flux and glucose oxidation has been overstated. We therefore have adjusted title, abstract and the overall narrative to present these data in a more nuanced manner.

Reviewer #1 (Remarks to the Author):

The authors have provided additional evidence to support the role of PanK4 in modulating muscle substrate utilization. While the additional experiments and data have substantially strengthened some of their conclusions, they also lead to additional questions. Even with the revisions, unfortunately, the mechanistic work remains preliminary and difficult to follow.

Additional comments below:

1. The authors have now nicely demonstrated the abundance of PanK4 across different skeletal muscle beds by Western blotting in new Fig 1F. It would be helpful if the quantification of Pank1 and 2 were also added to the main figure, so readers can easily see how the different isoforms are expression across muscle beds.

This is a valid point. We have added the quantification of PanK1 and 2 to the main Figure 1.

2. It would be helpful if the authors could include whether the analyses were done in human or mouse tissue (male or female), which muscle depot was used, and whether the experiments were conducted in fed/fasted states to aid in readability. Not all of this information is included within all figure legends and main text, making it difficult to compare findings across different experimental settings.

This is a relevant suggestion. We have included some of this information in the figure legends.

3. As mentioned in my initial (minor) comments, the SNP data is rather preliminary. How does the directionality of the variant correlate with the findings in mice? This might be better suited for the supplement.

Thanks for this fair suggestion. We have now moved this figure to the supplement.

4. I commend the authors for performing metabolic cage studies and realize that it is difficult to pick up changes in whole-body substrate utilization using this technique. Fasting/refeeding experiments can often give a bigger window to see differences in wholebody substrate utilization (especially during the initial transition to the fasting state-see PMID: 22517326, for example). From the RER plot, it looks like during the initial part of the fast the KO mice utilize more carbs (RER closer to 1)—this assumes the inset though

has the correct alignment of when the fast actually started. If the inset is correct- was carbohydrate utilization significantly increased during this time? Average activity is also increasing in the KO mice, which seems relevant to discuss in the results given the effect of PanK4 KO on exercise.

Thank you for your detailed feedback regarding the metabolic cage studies. As you suggested, we calculated carbohydrate and lipid utilization using the caloric stoichiometry of oxygen for lipid and carbohydrate oxidation, respectively (see data below). Overall, there are no significant differences in whole-body substrate utilization between genotypes. However, there sometimes “appears” to be a slight circadian misalignment in carbohydrate use in PanK4 mKO mice compared to WT mice. Regarding the initial fasting response, in the first two hours of the fasting challenge, there is a non-significant trend, as you correctly emphasize, of slightly increased carbohydrate use and decreased lipid use. However, this coincides with what seems to be increased activity during that time, so we are hesitant to emphasize these findings. Regarding the finding of increased activity of PanK4 mKO mice on chow diet, we now mention these data (p 5, ln 8-9).

Related to these data, we now also discuss the possibility that the impact of PanK4 might be fiber-type dependent. In fact, we have gone through the manuscript and highlighted the type of muscle that were tested to highlight the use of oxidative vs glycolytic muscle. It maybe that in oxidative muscle PanK4 is important for regulating FAOX, while in glycolytic muscles PanK4 controls glucose uptake. While speculative (and requiring future work) this could explain the moderate effect on whole-body substrate utilization; in addition, to the mouse being a more liver-centric organisms compared to humans.

5. Were levels of fasting/fed plasma glucose, NEFAs or BOHB altered in the muscle specific KO mice?

We, unfortunately, do not have plasma left from the fasting/fed experiment. We have previously assessed/included data on plasma glucose after 5-6 h fast (before GTT) (Fig 4a,b) and after 20 min of rest or exercise (Fig S7a). We have also previously measured/included data on NEFAs in glucose-injected PanK4 WT and PanK4 mKO mice (Fig S4d). BOHB levels are a good idea, thanks for that suggestion, and therefore we determined plasma BOHB in the terminal exercise study in which we observed the impaired glucose uptake during running in PanK4 mKO mice. For BOHB we observed no effects of genotype on plasma BOHB levels (data below). Of course, these data only provide a snapshot and in theory it remains possible that a higher output of BOHB by liver is matched by a higher uptake of BOHB in muscle. Thus, we consider these BOHB data preliminary for now and will not include them so that we can assess ketone body use in PanK4 mKO mice with more rigor in the future.

As presented, it is often unclear what groups are statistically significant in the figures. It would be helpful to always present the data in the same way (KO next to controls within fed or fasted state).

We have tried to make the figures clearer by following the reviewer's suggestion to compare WT and KO within specific conditions (e.g., fed or fasted; or basal or insulin).

The authors now nicely show that PanK4 regulates glucose uptake during exercise in vivo but there are no experiments showing that PanK4 regulates glucose utilization in vivo (only PDH

activity is measured *ex vivo* in KO mice and glucose uptake is only assessed in the O/E model; further pPDH is not changing *in vivo*). It is further still unclear how PanK4 regulates acetyl-CoA levels. Further, if glucose and fat utilization is down, what substrate is the muscle using?

Addressing insulin-stimulated *in vivo* glucose uptake (via e.g. clamp technique) is challenging and not feasible within the short time frame provided for these revisions. As the reviewer notes, we do show that PanK4 is required for muscle glucose uptake during running exercise, for *ex vivo* insulin-stimulated glucose uptake into glycolytic muscle, and that PanK4 overexpression is sufficient to increase glucose uptake into glycolytic muscle. Our aggregate data provide strong evidence that PanK4 regulates glucose uptake in muscle.

Regarding the PDH data, we have toned down the interpretation of these data considerably everywhere in the manuscript because we agree that there are too many unanswered questions related to these data.

Concerning how PanK4 regulates acetyl-CoA, the reviewer is correct that this remains unknown. But we hope the reviewer agrees that this is not low-hanging fruit. We point to unanswered question in the discussion (p 10, ln 18-28). To be fair, this question of why acetyl-CoA is regulated by PanK4 or where in the cell acetyl-CoA is elevated was also not addressed in the 2022 *Nature* publication¹.

Re the question what substrate(s) muscle is using (instead of glucose and FAs), we share the reviewer's enthusiasm for this topic. For now, we can only speculate. It is possible that there is an increased reliance on ketones, shorter-chain fatty acids (C10 or shorter) that enter mitochondria independently of the carnitine shuttle, or certain amino acids. We now discuss this open question (p 10-11, ln 29-34, 1-8). Interestingly, we now include new data showing that muscle glycogen stores are increased in PanK4 mKO mice (Fig S7i), which could explain why endurance capacity was normal in PanK4 mKO mice.

Overall, we acknowledge that there are still open questions, which to some degree should be expected for a new topic; however, we strongly believe that the work we have presented (especially pertaining to highly relevant metabolic endpoints) meets the standards for publication in *Nature Communications*.

Reviewer #2 (Remarks to the Author):

The investigators have performed a substantial amount of additional work and have provided several new data sets. While the manuscript has been improved, some of the most important gaps that had been identified previously were not sufficiently satisfied.

Major comments

1. Considering that the report concludes that PanK4 controls muscle oxidation of both fatty acid and glucose, reviewers asked how the loss of PanK4 in muscle affects energy expenditure, substrate utilization, and exercise performance at the whole-body level, as well as muscle PDH phosphorylation and PDK4 stability and abundance. To their credit, investigators produced and shared several new analyses to address these questions. The results of all those experiments and analyses were surprisingly negative. While negative data can be critically important for developing a clear picture of function and mechanism; in aggregate, the negative outcomes here raise strong doubt that loss of PANK4 results in diminished PDH flux, reduced glucose oxidation, and inefficient energy metabolism - as implied in the title, abstract, and narrative. There is simply no evidence to support these aspects of the working model.

Overall, the reviewer raises some fair points. However, we disagree with the notion that the lack of detectable differences at the whole-body level diminishes the relevance of our findings at the muscle level. Mice are more liver-centric organisms compared to humans. While skeletal muscle is the primary site of glucose disposal in humans, it is the liver in rodents. This is evident from the fact that the relative liver weight is approximately 2.5 times greater in rats and mice than in humans, and glycogen levels in human muscle are 20-fold higher than in mouse muscle. Moreover, in terms of whole-body physiological impact, the marked glucose intolerance observed should not be understated.

We do, however, agree that the link between PDH flux and glucose oxidation has been overstated. We therefore have adjusted title, abstract and the overall narrative to present these data in a more nuanced manner. The PDH findings are what led us towards investigating endpoints related to glucose homeostasis, which is how we now present these data, as an intriguing clue rather than a causal link.

While the reviewer is likely correct that glucose oxidation is unaltered, this makes it even more intriguing that insulin-stimulated glucose uptake in glycolytic muscle and exercise-induced glucose uptake into muscle overall is impaired. The inability to efficiently take up glucose in muscle is a key risk factor for developing type 2 diabetes in humans², which is why we think these data are highly relevant.

2. Figure 4d shows that insulin-stimulated 2DOG uptake is decreased in EDL, whereas basal levels were unchanged, suggesting that the KO EDL muscles exhibit diminished insulin action at the level of glucose uptake, without affecting insulin signaling. By contrast, neither basal nor insulin-stimulated 2DOG uptake were affected in Pank4KO soleus muscles. Also noteworthy, Figure 2 shows that KO soleus muscles had lower oxidation rates of ¹⁴C fatty acid when assayed *ex vivo*, but only when the muscles were stimulated to contract. Therefore, the impact of Pank4 on both glucose uptake and fat oxidation appears to depend on fiber type and context, even though acetyl CoA was elevated in all states tested. Similar assays using radio-labeled glucose could be performed in both soleus and EDL to determine if changes in glucose oxidation rates are indeed evident under these same conditions. In sum, the finding that loss of Pank4 results in diminished uptake of 2DOG into muscles under some (but not all) circumstances does not establish perturbed PDH flux and/or glucose oxidation. While the investigators' assumption might be correct, this claim remains entirely speculative based on the data provided. Also at odds with this suggestion is that acetyl CoA, malonyl CoA and citrate levels are elevated in mKO muscles, which typically occurs when PDH flux increases.

Thank you for these comments.

- 1) Correct, insulin-stimulated glucose uptake is impaired in EDL but not soleus muscle, possibly suggesting fiber-type dependent interactions since the soleus is a more oxidative muscle whereas the EDL is highly glycolytic. This theme of potential fiber-type dependent effects of Pank4 is something we discuss now.
- 2) It is incorrect that Figure 2 shows that FAOX is impaired in soleus only with contractions. Instead these data show that there is a main effect of genotype: FAOX is overall lower in Pank4 mKO independent of stimulation (basal vs contraction). We changed the figure design to make this clearer to the reader.
- 3) We share the reviewer's enthusiasm for investigating fiber-type specific regulation of muscle substrate metabolism by Pank4, but this requires a future systematic approach that is beyond a reasonable time frame for a revision. Regarding the point of acetyl-CoA being up indiscriminately in different states and muscle. That is true, but it is possible that

the cellular impact is fiber-dependent. Mitochondria content among several factors could play a role in how a cell or a muscle fiber deals with elevated levels of metabolites such as acetyl-CoA.

- 4) The reviewer makes a fair point when he/she writes “the finding that loss of Pank4 results in diminished uptake of 2DOG into muscles under some (but not all) circumstances does not establish perturbed PDH flux and/or glucose oxidation”. Accordingly, we have tempered down this narrative throughout the manuscript and focus on glucose uptake rather than oxidation.
 - 5) The last point regarding “...acetyl CoA, malonyl CoA and citrate levels are elevated in mKO muscles, which typically occurs when PDH flux increases” is of course fair however as we learned and as the reviewer is aware there are many ways for the cell to produce acetyl-CoA (not to mention the many metabolic pathways that consume acetyl-CoA) and several of these would be uncoupled from PDH flux. So, it is not that simple and we try to highlight this complexity in the discussion (p 10, ln 18-28).
3. The investigators made a strong and commendable effort to assay malonyl CoA levels. When taken together with the other findings, these new data do provide solid evidence that KO of Pank4 results in diminished fat oxidation, likely due in part to malonylCoA-mediated inhibition of CPT1a.

Thanks for acknowledging our efforts regarding malonyl-CoA. It was indeed a struggle, but we are excited to have this assay established now.

4. Considering that KO of Pank4 did not affect free CoA levels and markedly increased acetyl CoA, it remains unclear whether or not this protein actually regulates CoA synthesis and why acetyl CoA increases. The results implying that Pank4 binds to BDH1 are quite intriguing; however, these findings are impossible to interpret without further assessment of muscle ketone metabolism.

We show that free CoA levels remain unchanged, but we cannot rule out the possibility that total CoA, which includes free CoA and all CoA-containing metabolites, may be altered. We now mention this limitation (p 9, ln 8-12) along with the caveat that our CoA measurements provide only a singular snapshot.

The reason for the increase in acetyl-CoA remains unclear. Our data suggest that major acetyl-CoA-producing pathways, such as carbohydrate, lipid, and protein catabolism, are not responsible. However, more research is needed in this area. It's also possible that acetyl-CoA-consuming pathways are underutilized; for example, alterations in polyamine metabolism, which has been linked to changes in acetyl-CoA content³, could be a factor. Clearly, the question of why acetyl-CoA is elevated is complex, and fully addressing it may go beyond the scope of this single publication. To be fair, this question of how PanK4 regulates acetyl-CoA was also not addressed by the 2022 *Nature* publication¹. Similarly, it was not investigated where in the cell acetyl-CoA was altered.

Regarding BDH1, the reviewer makes a valid point. We now present these data as neutrally as possible, though we believe it is an interesting observation worth mentioning to the reader.

5. The report shows that overexpression of Pank4 lowers acetyl CoA and increases glucose uptake, but does not provide information on how this manipulation affected fat oxidation and glucose oxidation.

With students graduating and funding cycles ending, we currently lack the personnel for virus production, injections, and other tasks necessary to repeat these experiments in a reasonable timeframe. Whether these data (as they are now) should be included may ultimately be an editorial decision. However, we still believe the findings are highly intriguing, as they show that PanK4 is sufficient to increase glucose uptake into muscle, which complements showing impaired glucose uptake when PanK4 is missing. Especially, with the emphasis now being on glucose uptake rather than oxidation.

Minor comments

The title states that PanK4 controls efficient substrate use. What is meant by “efficient”? What is the evidence of inefficiency?

We have changed the title to “Pantothenate Kinase 4 controls skeletal muscle substrate metabolism”.

If muscle glucose uptake is diminished in KO mice during exercise, but exercise tolerance and performance are unaffected; how are the mice supporting muscle contraction? Is there any evidence that use of ketones and/or muscle glycogen are increased in KO mice? Was muscle lactate production different?

Following the reviewer’s suggestion, we measured muscle glycogen in the terminal exercise glucose uptake study. Interestingly, overall glycogen levels are increased in PanK4 mKO mice compared to control WT mice. This could explain why PanK4 mice have similar endurance capacity than WT mice. Notably, it has also been proposed that glycogen levels can modulate insulin-stimulated glucose uptake, with high glycogen levels possibly decreasing insulin-controlled glucose uptake (compared to lower glycogen levels) particular in glycolytic muscles⁴. We include these glycogen data now in Fig S7i.

We were unable to measure muscle lactate production, however we measured blood lactate at rest and during exercise and observed the expected increase with exercise. There was no genotype difference for blood lactate levels. We include these data now in Fig S7b.

We also measured BOHB levels, but found no difference due to exercise (likely too short) and genotype. Please see data here:

Of course, these data only provide a snapshot and in theory it remains possible that a higher output of BOHB by liver is matched by a higher uptake of BOHB in muscle. Thus, we consider these BOHB data preliminary for now and will not include them so that we can assess ketone body use in PanK4 mKO mice with more rigor in the future.

Considering that Pank4 localizes to the IMS and appears to elevate the malonyl CoA pool that regulates CPT1 activity, have the authors considered the possibility that Pank4 is regulating the acetyl CoA pool that resides outside the mitochondria, rather than the matrix pool?

We have considered this and mention this in the discussion.

In the instances when differences between groups do not reach statistical significance, results should be discussed as such (e.g. PDH activity, basal fat oxidation).

We have done this now for PDH, highlighting that it is a trend for an effect. Regarding basal fat oxidation beware our answer above: "...these data show that there is a main effect of genotype: FAOX is overall lower in PanK4 mKO independent of stimulation (basal vs contraction)."

Reviewer #1 (Remarks to the Author):

I commend the authors for performing additional work to answer key questions raised by both reviewers and toning down their conclusions during this revision process. While I acknowledge the value in identifying Pank4 as an exercise target and believe the work is important and novel, unfortunately the major concerns that were raised by both reviewers are still not sufficiently addressed. The lack of consistency in terms of the whole-body substrate utilization and mechanistic work (circulating substrates, glucose uptake, metabolomics, PDH activity) with different stimuli (exercise, contraction, HFD) and muscle fiber types (oxidative vs glycolytic) makes it hard to draw conclusions about the mechanism(s) by which Pank4 alters substrate utilization, and unfortunately, lessens my enthusiasm for this work.

Reply: We thank the reviewer for their thoughtful assessment and appreciate their recognition of the value in identifying Pank4 as an important and novel exercise target. While we acknowledge the reviewer's point that there are unresolved questions at this stage, we believe this is characteristic of a developing research field. Our data demonstrating that Pank4 is an exercise target, its abundance in muscle, the effects of whole-body KO on body size and IGF-1, its role in regulating acetyl-CoA and malonyl-CoA in muscle, its impact on glucose and lipid metabolism, and its ability to increase glucose uptake are all significant findings we believe are highly relevant for the scientific community.

Reviewer #2 (Remarks to the Author):

The revised title and narrative are now much better aligned with the data presented. The work is significant and provides important new insights into the role of Pank4 in regulating muscle metabolism. Overall, the findings are novel and provocative, and likely to attract strong interest from the metabolic research community.

Reply: We appreciate the reviewer's thoughtful evaluation and their valuable suggestions for improving the work. We are pleased that the reviewer finds the final manuscript engaging and impactful.